# *Plasmodium*-infected erythrocytes induce secretion of IGFBP7 to form type II rosettes and escape phagocytosis

Wenn-Chyau Lee[1], Bruce Russell[2], Radoslaw Mikolaj Sobota[3,4], Khairunnisa Ghaffar[1], Shanshan W Howland[1], Zi Xin Wong[1], Alexander G Maier[5], Dominique Dorin-Semblat[6,7], Subhra Biswas[1], Benoit Gamain[6,7], Yee-Ling Lau[8], Benoit Malleret[1,9], Cindy Chu[10,11], François Nosten[10,11], Laurent Renia[1]*

[1]Singapore Immunology Network (SIgN), Agency for Science, Technology and Research (A*STAR), Singapore, Singapore; [2]Department of Microbiology and Immunology, University of Otago, Dunedin, New Zealand; [3]Systems Structural Biology Group, Institute of Molecular and Cell Biology (IMCB), Agency for Science, Technology and Research (A*STAR), Singapore, Singapore; [4]Institute of Medical Biology (IMB) Agency for Science, Technology and Research (A*STAR), Singapore, Singapore; [5]Biomedical Sciences and Biochemistry, Research School of Biology, Australian National University, Canberra, Australia; [6]Université de Paris, Biologie Intégrée du Globule Rouge, UMR_S1134, BIGR, INSERM, Paris, France; [7]Institut National de la Transfusion Sanguine, Paris, France; [8]Department of Parasitology, Faculty of Medicine, University of Malaya, Kuala Lumpur, Malaysia; [9]Department of Microbiology and Immunology, Yong Loo Lin School of Medicine, National University of Singapore, Singapore, Singapore; [10]Shoklo Malaria Research Unit, Mahidol-Oxford Tropical Medicine Research Unit, Faculty of Tropical Medicine, Mahidol University, Mae Sot, Thailand; [11]Centre for Tropical Medicine, Nuffield Department of Medicine, University of Oxford, Oxford, United Kingdom

*For correspondence: renia_laurent@immunol.a-star.edu.sg

**Competing interests:** The authors declare that no competing interests exist.

**Abstract** In malaria, rosetting is described as a phenomenon where an infected erythrocyte (IRBC) is attached to uninfected erythrocytes (URBC). In some studies, rosetting has been associated with malaria pathogenesis. Here, we have identified a new type of rosetting. Using a step-by-step approach, we identified IGFBP7, a protein secreted by monocytes in response to parasite stimulation, as a rosette-stimulator for *Plasmodium falciparum*- and *P. vivax*-IRBC. IGFBP7-mediated rosette-stimulation was rapid yet reversible. Unlike type I rosetting that involves direct interaction of rosetting ligands on IRBC and receptors on URBC, the IGFBP7-mediated, type II rosetting requires two additional serum factors, namely von Willebrand factor and thrombospondin-1. These two factors interact with IGFBP7 to mediate rosette formation by the IRBC. Importantly, the IGFBP7-induced type II rosetting hampers phagocytosis of IRBC by host phagocytes.

## Introduction

Along its intraerythrocytic development, *Plasmodium* spp. modifies the infected erythrocyte (IRBC) rheology (increased rigidity for *P. falciparum* IRBC, reduced rigidity but increased fragility for *P. vivax* IRBC). Such alteration increases the susceptibility of IRBC to splenic clearance (*Handayani et al., 2009*; *Chotivanich et al., 2002*); however, the parasites have developed escape strategies to avoid splenic elimination. For instance, *P. falciparum* expresses adhesins on the IRBC that mediate

**eLife digest** Malaria is a life-threatening disease transmitted by mosquitoes infected with *Plasmodium* parasites. Part of the parasite life cycle happens inside human red blood cells. The surface of an infected red blood cell is coated with parasite proteins, which attract the attention of white blood cells called monocytes. These immune cells circulate in the bloodstream and use a process called phagocytosis to essentially 'eat' any infected cells they encounter. However, the monocytes cannot always reach the infected cells.

Some of the proteins made by the parasites make the infected red blood cells stickier than normal. This allows the infected red blood cells to surround themselves in a protective cage of uninfected red blood cells. Known as "rosettes" because of their flower-like shape, these cages seem to protect the infected cells from attack by the immune system. Lee et al. noticed that adding white blood cells to parasite-infected red blood cells made them clump together more, but it was unclear exactly how and why this happened.

To find out, Lee et al. took fluid from around monocytes grown in the laboratory and added it to red blood cells infected with *Plasmodium* parasites. This made the cells clump together, suggesting that something in the fluid may potentially be alerting the parasites to impending immune attack. The fluid contained almost 700 different molecules, and Lee et al. narrowed down their investigations to the five most likely candidates. Interfering with the activities of these five proteins revealed that one – a protein IGFBP7 – not only alerted the parasites but also helped them to form the rosettes. It turns out that the parasites appear to use IGFBP7 like a bridge, linking it to two other human proteins to stick red blood cells together. Once the rosettes had formed, the monocytes were unable to eat the infected blood cells by themselves. Instead several monocytes had to work together as a team to consume the whole rosette.

Further research is now needed to shed light on this interaction between malaria parasites and human cells. Such research would be particularly relevant in the clinical setting, since some previous studies has linked the forming of rosettes to the severity of disease for malaria.

adhesion to the endothelial cells, resulting in deep microvasculature sequestration (*Sherman et al., 2003*). Additionally, an IRBC can bind directly to uninfected red blood cells (URBC) to form a 'rosette' (*Russell and Cooke, 2016*). This rosetting phenomenon has been described in all human malaria parasites (*Angus et al., 1996*); however, the functional importance of rosetting remains ambiguous (*Clough et al., 1998*; *Wahlgren et al., 1989*). While the supposed role of rosetting in facilitation of merozoite invasion of URBC is unlikely; recent studies show that rosette formation may have a role in parasite immune-evasion (*Clough et al., 1998*; *Deans and Rowe, 2006*; *Lee et al., 2014*; *Zhang et al., 2016*; *Moll et al., 2015*; *Lee et al., 2019*). Theoretically, the masking of rosetting IRBC with URBC may hamper IRBC recognition and therefore their clearance by the host immune system (*Moll et al., 2015*). Notably, rosetting has been associated (in some but not all studies) with disease severity (*Treutiger et al., 1992*; *Carlson et al., 1994*; *Rowe et al., 1995*; *Doumbo et al., 2009*; *al-Yaman et al., 1995*; *Ho et al., 1991*). Here, we observed that the addition of leukocytes increased the rosetting rates of various *P. falciparum* and *P. vivax* isolates. We next demonstrated that IRBC stimulated monocytes to secrete products capable of stimulating rosetting, the most important being insulin growth factor binding protein 7 (IGFBP7). We further showed that IGFBP7-mediated rosetting was different from the previously described rosetting (defined here as type I rosetting), where it (we refer to this as type II rosetting) required additional serum factors to occur, in addition to the interaction between the parasite-derived ligand on IRBC surface and the receptor on the surface of URBC. Functionally, we observed that the IGFBP7-mediated type II rosetting reduced phagocytosis by monocytes, and therefore defined a new escape mechanism for the malaria parasites.

## Results

The key resources table is available as *Supplementary file 1*.

## Effects of human leukocytes on rosetting

*P. falciparum*- and *P. vivax*-IRBC formed rosettes (*Figure 1A*, top panel) in the presence of 20% autologous human serum and the extent of rosetting at baseline is variable depending on the parasite isolates (*Lee et al., 2014*). When autologous blood leukocytes were incubated with clinical isolates, the rosetting rate increased by 10–40% depending on individual isolates for both parasite species (*Figure 1B*). This effect was mediated primarily by monocytes (*Figure 1C*). To exclude donor variability as a confounding factor, we repeated the experiment with the THP-1 cell line, which was derived from the peripheral circulation of an acute monocytic leukaemia patient (*Auwerx, 1991*). The rosette-stimulation mediated by THP-1 was similar to that of peripheral monocytes (*Figure 1D*). Addition of undifferentiated THP-1 (UT) and macrophage-like THP-1 (MT) increased rosetting rates in a dose-dependent manner. MT were more potent than UT, where the difference increased with the number of cells added (*Figure 1E*).

## Effect of THP-1 culture supernatants and supernatant fractionation

Culture supernatants (CS) of both THP-1 cell types showed similar rosette-stimulation effects (*Figure 1F*), with CSMT exerting a higher degree of rosette-stimulation than CSUT. Fractionation of CSMT into aqueous and lipid fractions revealed that the rosette-stimulating factors were in the aqueous fraction (*Figure 2A*). Subsequently, we further fractionated the aqueous fraction into high and low molecular weight sub-fractions (with a cut-off of 30 kDa). Both aqueous sub-fractions induced rosetting of *P. falciparum* and *P. vivax* (*Figure 2B and C*), demonstrating that the rosetting stimulation was mediated by multiple secreted hydrophilic factors, predominantly of sizes $\leq$ 30 kDa (particularly for *P. falciparum*). Heating of CSMT's $\leq$ 30 kDa aqueous fraction at 56°C for 1 h did not completely abolish its rosette-stimulating effect (*Figure 2D*), indicating the presence of heat-stable factors. To further investigate the potential stimulating factors, mass spectrometry analysis was performed on this fraction. We identified 694 proteins (*Supplementary file 2*), of which complement factor D (CFD), insulin-like growth factor binding protein 7 (IGFBP7), nidogen 1 (NID1), hyaluronan-binding protein 2 (HABP2), and periostin (OSF2) were shortlisted (*Supplementary file 3*). Apart from having high scores in the mass spectrometry analysis (*Supplementary file 2*), these molecules were selected because they are secreted proteins and possess cell adhesion-related properties associated with pathological changes such as vascular injuries, inflammation, cancer development, and leukocyte recruitment (*UniProt Consortium, 2015*; *St Croix et al., 2000*; *Mambetsariev et al., 2010*; *Ulazzi et al., 2007*; *Schwanekamp et al., 2016*; *Lindner et al., 2005*; *Markiewski and Lambris, 2007*).

## Antibody neutralization assay

Antibody neutralization of IGFBP7 significantly reduced (by ~40%) the rosette-stimulation by CSMT for both parasite species (*Figure 3A*). Rosette-stimulation by CSMT was also reduced by anti-CFD (*Figure 3B*) and anti-OSF2 (*Figure 3C*) antibodies, albeit to a lesser extent. Antibodies against NID1 (*Figure 3D*) and HABP2 (*Figure 3E*) had no effect on CSMT-mediated rosette-stimulation. As anti-IGFBP7 had the largest inhibitory effect, further experiments focused on IGFBP7.

## Effect of IGFBP7 on rosetting

Addition of human recombinant IGFBP7 to leukocyte-free parasite culture stimulated rosette formation in a dose-dependent and satiable manner for *P. falciparum* and *P. vivax* isolates, reaching a plateau at 100 ng/ml (*Figure 4A and B*). For *P. falciparum*, no significant difference in rosette-stimulatory effect between the CSMT and IGFBP7 was found (*Figure 4C*). On the contrary, *P. vivax* rosette-stimulation by CSMT was significantly higher than that by recombinant IGFBP7 alone (*Figure 4D*). The rosette-stimulating capacity of IGFBP7 was abolished by heat denaturation at 95°C for 1 h (*Figure 4E and F*), thus eliminating the possibility that the effect was caused by a non-protein contaminant in the recombinant protein preparation. Furthermore, the IGFBP7 binding-inducing effect was only observed when URBC and IRBC were present in the culture. Incubation of URBC alone with IGFBP7 did not induce clustering effect on the cells (*Figure 4—figure supplement 1*; top panel). IGFBP7 required a minimum of 15 min to significantly stimulate rosetting. However, the rosette-stimulating effect did not significantly increase further afterwards (*Figure 4G*). Importantly, IGFBP7-mediated rosetting was a reversible event, where removal of the protein from the system

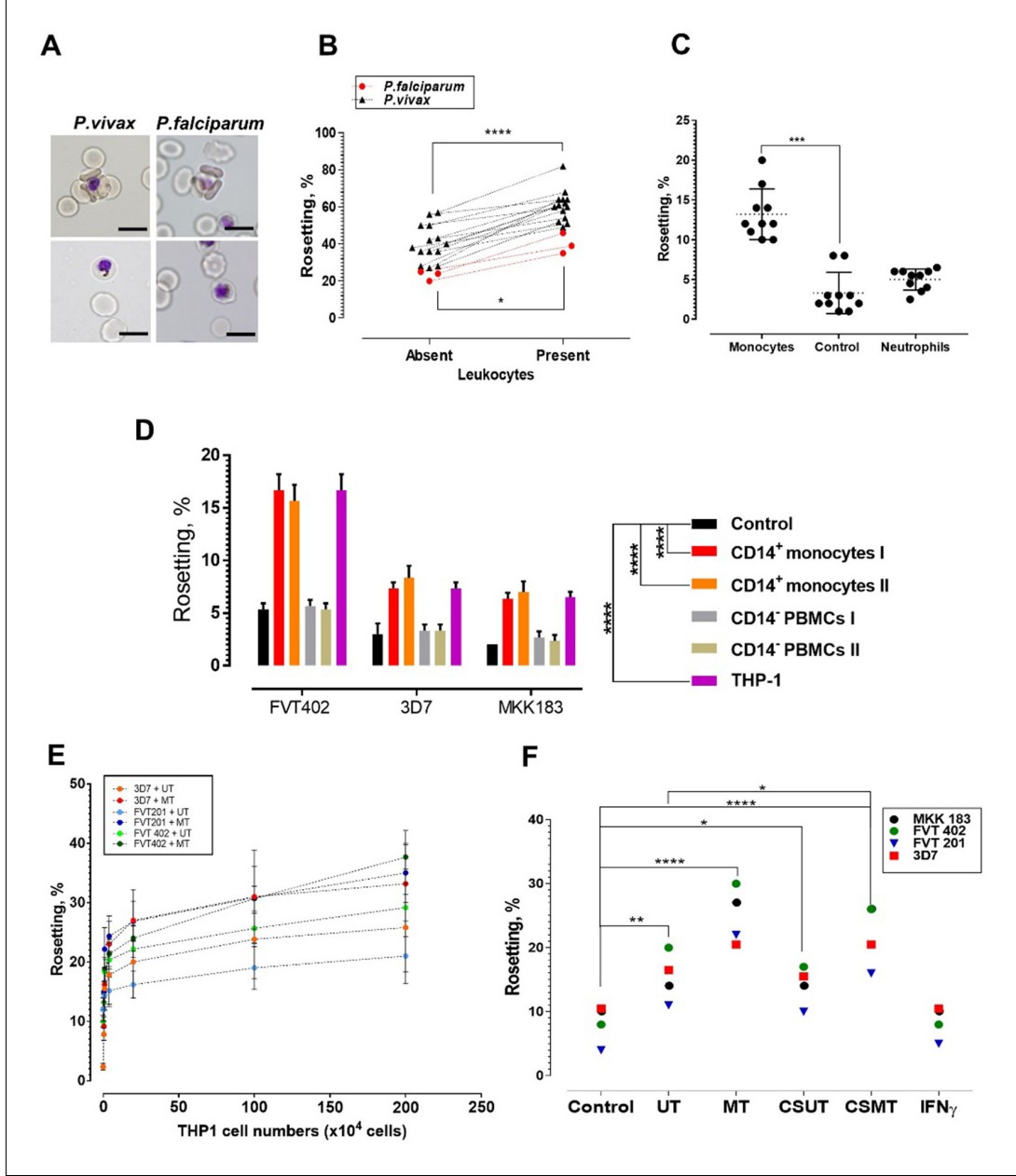

**Figure 1.** Deciphering the leukocyte subpopulation that influences rosetting. (**A**) Rosette (above) and non-rosetting (bottom) IRBC in culture medium for respective species. Scale bars represent 10 μm; oil immersion (1000×) magnification. Pictures taken with microscope camera Olympus DP21 on light microscope Olympus BX43. (**B**) Effect of autologous leukocytes on rosetting [*P. falciparum* (*Pf*, n = 3) (paired t-test p=0.0115) and *P. vivax* (*Pv*, n = 14) (paired t-test p<0.0001)]. (**C**) Comparison of effects of monocytes and neutrophils from different healthy individuals (n = 3) on *Pf* (n = 4) rosetting, Friedman test with Dunn's multiple comparison test p=0.0003 for comparison between control and monocytes; p=0.7422 between control and neutrophils. (**D**) Rosetting rates of lab-adapted *Pf* lines under different experimental conditions (co-incubated with $1 \times 10^5$ purified CD14+ peripheral monocytes and CD14− PBMC fractions from two healthy individuals, $1 \times 10^5$ THP-1s, and control without co-incubation with WBCs). Means and SD of triplicate (three biological replicates) experiments shown. Two-way ANOVA with Tukey's multiple comparison test: adjusted p<0.0001 in control vs. both CD14+ and control vs. THP-1 for all parasite lines. No significant difference found in THP-1 vs. CD14+ (p>0.5 for all parasite lines), and control vs. CD14− (p>0.5 for all parasite lines). (**E**) Plot showing changes of rosetting rates of three laboratory-adapted *P. falciparum* lines when incubated with different numbers of UT and MT separately. Means and SD of triplicate experiments shown. (**F**) Rosetting of lab-adapted *Pf* lines under different experiment conditions [control, with UT, MT, and with culture supernatant of UT (CSUT) and MT (CSMT)]. One-way ANOVA with Tukey's multiple

*Figure 1 continued on next page*

*Figure 1 continued*

comparison test: control vs. UT: p=0.0072; control vs. MT: p<0.0001; control vs. CSUT: p=0.0293; control vs. CSMT: p<0.0001; control vs. IFNγ: p>0.9999; UT vs. CSUT: p=0.1411; MT vs. CSMT: p=0.1397. UT vs. CSMT: p=0.033; df = 3. *p<0.05, **p<0.01, ***p<0.001, ****p<0.0001.
The online version of this article includes the following source data for figure 1:

**Source data 1.** Raw data (rosetting rates, %) for the data set presented in the bar graph (*Figure 1D*).

reverted the rosetting rates to their baseline values (rosetting rates recorded prior to IGFBP7 exposure) as fast as 15 min post-protein removal (*Figure 4H*).

## Parasite-derived rosetting ligands essential to IGFBP7-mediated rosetting

Trypsinization of IRBC abrogated the rosette-stimulating effect of IGFBP7 (*Figure 5A*), suggesting the involvement of parasite-derived proteins expressed on the surface of IRBC. *P. falciparum* expressed different surface antigens including rosetting ligands such as the *P. falciparum* erythrocyte membrane proteins 1 (PfEMP1), STEVOR, and RIFIN (*Chan et al., 2014*). The sensitivity to low level of trypsin treatment (10 µg/ml) used here suggested that PfEMP1, rather than STEVOR or RIFIN, was likely the adhesin involved (*Kyes et al., 1999*; *Niang et al., 2014*). We further validated this hypothesis with use of the *P. falciparum* SBP1-KO-CS2 line, in which the skeleton binding protein 1 (SBP1) gene was knocked out. SBP1 transports PfEMP1 from the parasitophorous vacuole to the Maurer's cleft for subsequent assembly and export to the surface of IRBC. Therefore, the SBP1-KO-CS2 line is unable to express PfEMP1 on the surface of the IRBC. However, knock out of this gene does not affect the surface expression of STEVOR and RIFIN (*Maier et al., 2007*; *Chan et al., 2016*). IGFBP7 increased the rosetting rate of CS2 wild-type parasite but had no effect on SBP1-KO-CS2 rosetting (*Figure 5B*). Subsequently, we used two other clones of the *P. falciparum* NF54 line, NF54 VAR2CSA_WT clone and the mutant clone, NF54_T934D, whose PfEMP1 variant VAR2CSA is not exported onto the surface of the IRBCs (*Dorin-Semblat et al., 2019*). The rosetting machinery of NF54_VAR2CSA_WT responded positively to the presence of IGFBP7. On the other hand, the rosetting rates of NF54_T934D clone (lacking PfEMP1 on the surface of IRBC) were not significantly altered by IGFBP7 (*Figure 5C*). Of note, IRBCs with surface expression of PfEMP1 variant VAR2CSA (which include the CS2_WT and NF54_VAR2CSA_WT) did not form many rosettes, which was in parallel with earlier report (*Rogerson et al., 2000*). In addition, the rosettes formed by NF54_VAR2C-SA_WT (*Figure 5C*, inset i) and NF54_T934D (*Figure 5C*, inset ii) were small.

Expression of parasite-derived, IRBC-surface proteins such as rosetting ligands is sequential and parasite stage-specific. For example, PfEMP1 is the first rosetting ligand to be expressed on the surface of IRBC (as early as late ring stage), followed by RIFIN, and finally STEVOR (which are at the much more mature stages) (*Kyes et al., 2000*; *Lavazec et al., 2007*; *Kaviratne et al., 2002*; *Bachmann et al., 2012*; *Niang et al., 2009*). In other words, PfEMP1 is the only rosetting ligand available on the surface of late ring-IRBCs. We performed the IGFBP7-rosetting assessment on the late ring stages (~hour 16–26) of a laboratory-adapted clinical isolate (nine replicates across three different cycles) and found that IGFBP7 significantly increased the rates of rosette formation (*Figure 5D*) by the late ring stages (*Figure 5E*). Taken together, these results suggest strongly that PfEMP1 is essential for IGFBP7-mediated rosetting in *P. falciparum*. On the other hand, we could not identify which *P. vivax* proteins were involved in IGFBP7-mediated rosettes as the *P. vivax*-IRBC membrane-associated proteins have yet to be fully characterized.

## Host-derived rosetting receptors essential to IGFBP7-mediated rosetting

IGFBP7 has a heparin binding domain. We hypothesized that it might be involved in the rosetting effect. When URBC treated with either heparinase I (*Figure 6A*) or heparinase III (*Figure 6B*) were mixed with purified *P. falciparum*- or *P. vivax*- IRBC, IGFBP7 did not induce rosetting. Heparinases also affected the size of rosettes (numbers of URBC in a rosette) (*Figure 6C*, green arrows). Notably, addition of IGFBP7 induced autoagglutination-like clustering of IRBC (which were not enzyme-

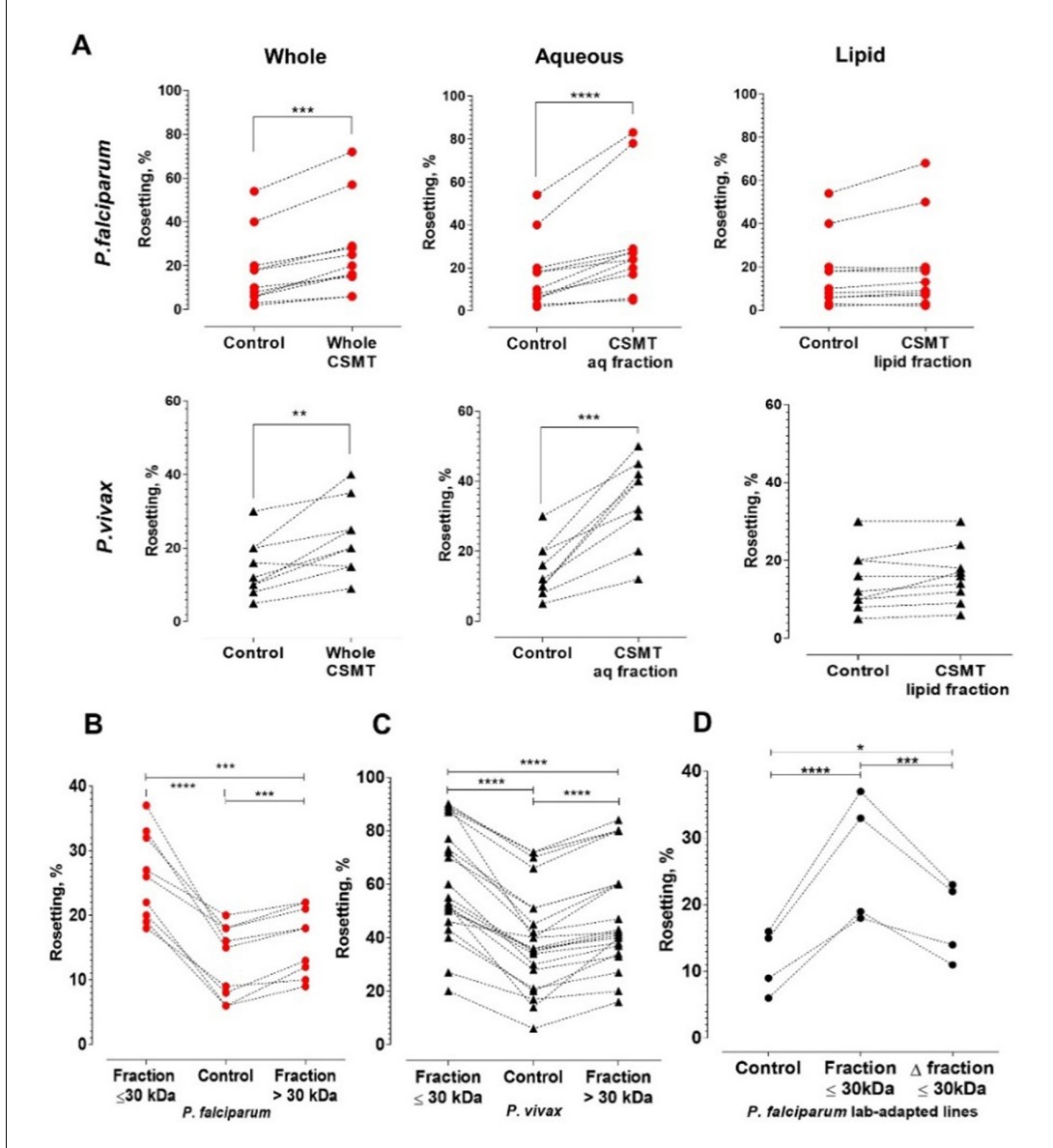

**Figure 2.** Characterization of the secreted rosette-stimulating factors. (A) Rosetting of clinical isolates (*Pf*: n = 11, *Pv*: n = 9) post-incubation with whole CSMT (paired t-test *Pf*: p=0.0001; *Pv*: p=0.0017) and its aqueous (*Pf*: p<0.0001; *Pv*: p=0.0002) and lipid (*Pf*: p=0.0954; *Pv*: p=0.0905) fractions. (B, C) *Pf* (n = 9) and *Pv* (n = 22) rosetting post-incubation with CMST aqueous fractions of different sizes. From one-way ANOVA with Tukey's test, both smaller-size (*Pf*: p<0.0001; *Pv*: p<0.0001) and larger-size (*Pf*: p=0.0007; *Pv*: p<0.0001) fractions significantly stimulated rosetting, with the smaller-size fraction exerting higher rosette-stimulation than the larger-size fraction (*Pf*: p=0.0005; *Pv*: p<0.0001). (D) Effect of heating on rosette-stimulation by the CMST aqueous ≤30 kDa fraction. The experiment was conducted with laboratory-adapted *P. falciparum* (n = 4). One-way ANOVA with Tukey's test: the unheated (p<0.0001) and heat-denatured (Δ) fractions (p=0.0236) stimulated rosetting, with the unheated fraction exerting higher stimulation than the heated fraction (p=0.0009). *p<0.05, **p<0.01, ***p<0.001, ****p<0.0001.

treated) (*Figure 6C*, red arrow). IRBC-autoagglutination was absent in the control groups (untreated and enzyme-treated groups without addition of IGFBP7) (data not shown).

Complement receptor 1 (CR1) expressed on URBC is a receptor for PfEMP1 in *P. falciparum* rosetting (*Rowe et al., 1997*). However, it did not play a significant role in IGFBP7-mediated rosetting for both parasite species. Anti-CR1 mAb reduced *P. falciparum* rosetting rates in the absence of

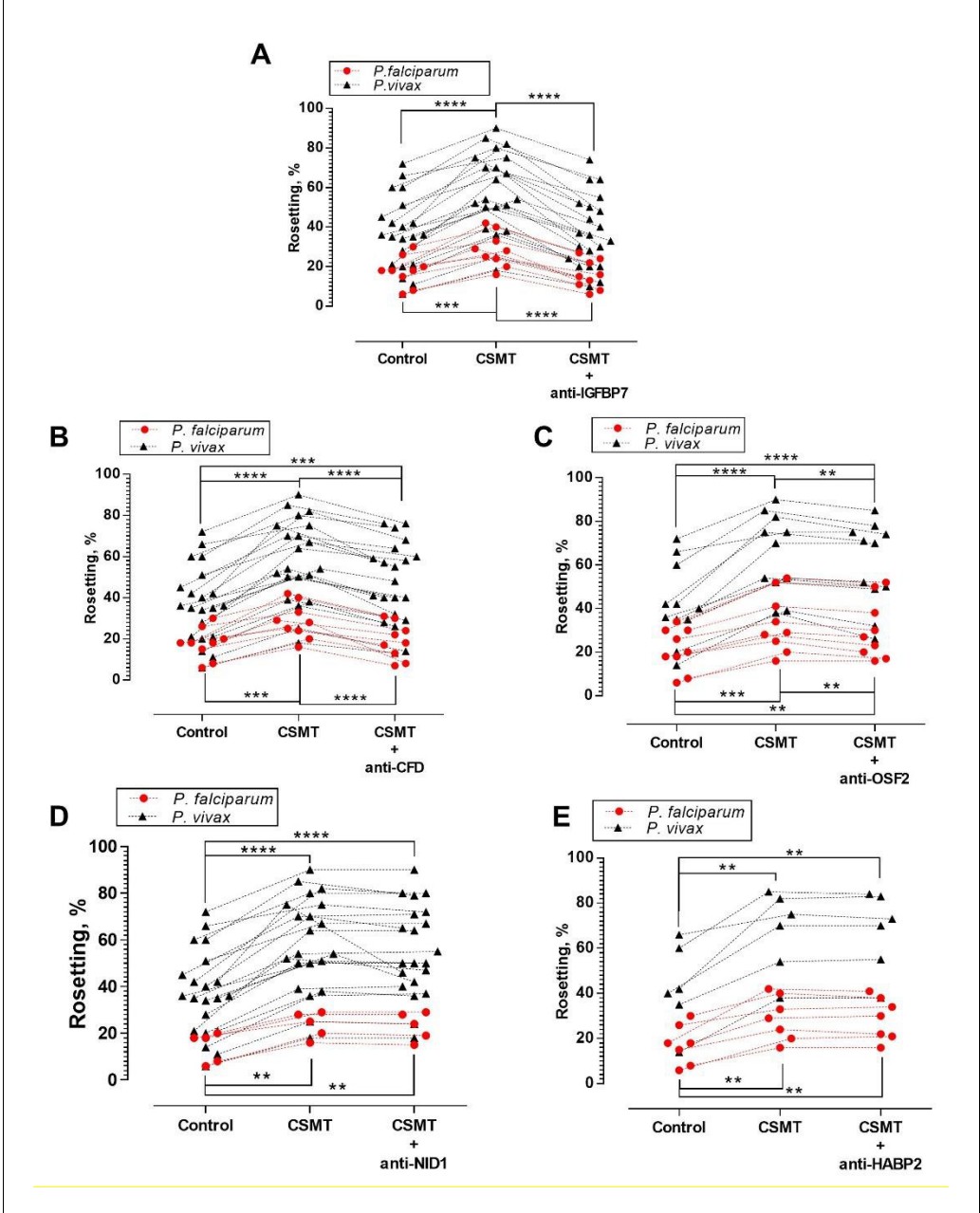

**Figure 3.** Antibody blocking assay. One-way ANOVA with Tukey's test was conducted to compare the rosetting rates between the control, CSMT and CSMT + antibodies against proteins of interest. (**A**) Rosetting rates were significantly increased by CSMT [*Pf* (n = 9): p=0.0008; *Pv* (n = 21): p<0.0001]. Anti-IGFBP7 significantly reduced the CSMT-mediated rosette-stimulation [*Pf*: p<0.0001; *Pv*: p<0.0001]. No significant difference was found between the control and CSMT + anti-IGFBP7 groups [*Pf*: p=0.5498; *Pv*: p=0.8724]. (**B**) Significant rosette-stimulation was found between control and CSMT [*Pf* (n = 9): p=0.0008; *Pv* (n = 21): p<0.0001]. Anti-CFD significantly reduced CSMT-mediated rosette-stimulation [*Pf*: p=0.0003; *Pv*: p<0.0001]. A significant difference was detected between control and CSMT + anti-CFD groups for *Pv* (p=0.0002), but not in *Pf* (p=0.8494). (**C**) Rosetting rates were significantly increased by CSMT [*Pf* (n = 9): p=0.0004; *Pv* (n = 11): p<0.0001]. Anti-OSF2 significantly reduced the CSMT-mediated rosette-stimulation [*Pf*: p=0.0017; *Pv*: p=0.0038]. A significant difference was detected between control and CSMT + anti-OSF2 groups for *Pf* (p=0.0058) and *Pv* (p<0.0001). (**D**) Rosetting rates were significantly increased by CSMT [*Pf* (n = 5): p=0.0038; *Pv* (n = 21): p<0.0001]. Anti-NID1 did not significantly alter CSMT-mediated rosette-stimulation [*Pf*: p=0.1432; *Pv*: p=0.1369]. A significant difference was found between control and CSMT + anti-NID1[*Pf*: p=0.0051; *Pv*: p<0.0001]. (**E**) Rosetting rates were significantly increased by CSMT [*Pf* (n = 7): p=0.0040; *Pv* (n = 6): p=0.0051]. Anti-HABP2 did not significantly alter CSMT-mediated rosette-stimulation [*Pf*:

*Figure 3 continued on next page*

*Figure 3 continued*
p=0.8514; *Pv*: p=0.9358]. A significant difference was found between control and CSMT + anti-HABP2 [*Pf*: p=0.0045; *Pv*: p=0.0073]. Of note, anti-human NID1 IgG was of the same subclass and raised in the same animal species as the antibodies used against IGFBP7, CFD, and OSF2 (*Supplementary file 1*). Hence, it also acted as a negative control in this set of experiments .* p<0.05, **p<0.01, ***p<0.001, ****p<0.0001.

IGFBP7, confirming that this molecule is involved in the direct interaction between IRBC and URBC (*Figure 6D*). However, anti-CR1 mAb had an insignificant effect on IGFBP7-induced rosetting in *P. falciparum* (*Figure 6D*). For *P. vivax*, anti-CR1 antibody did not inhibit rosetting, with or without addition of IGFBP7 to the culture (*Figure 6E*). Likewise, For *P. falciparum* and *P. vivax* (*Figure 6F and G*), ABO blood groups did not play significant roles in IGFBP7-mediated rosetting, as well as rosetting mediated by other CSMT-derived rosette-stimulators (*Figure 6H and I*).

## Serum-derived co-factors in IGFBP7-mediated rosetting

All the experiments described above were performed using 20% human serum-enriched medium. However, serum filtration with a 0.45 µm filter abolished IGFBP7-induced rosetting (*Figure 7A*). These results indicated that other large-sized serum-derived protein aggregates or multimers might be needed for the IGFBP7-mediated rosetting effect. We hypothesized that von Willebrand factor (VWF) may be involved as VWF has been reported to absorb onto plasma-exposed surfaces (*Grinnell and Phan, 1983*; *Mannhalter, 1993*). Addition of anti-VWF antibody (25 µg/ml) did not significantly alter the baseline rosetting rates (*Figure 7B*). However, the presence of anti-VWF antibody prevented IGFBP7 from exerting its rosette-stimulatory effect. The specificity of rosette-inhibition by the antibodies was validated with experiments using antibody isotype controls at the same working concentration (25 µg/ml) (*Figure 7—figure supplement 1A*).

When we reduced the medium's serum enrichment to only 2%, IGFBP7 could not increase rosetting rates (*Figure 7C*). The protein could only exert its rosette-stimulatory effect in 2% serum-enriched medium when VWF was added. Importantly, VWF by itself did not stimulate rosetting (*Figure 7C*), disputing the possibility of this phenomenon as a non-specific adhesive effect of VWF and indicating that it is a co-factor of the IGFBP7-mediated rosetting.

When the media's serum enrichment was replaced with Albumax II (a serum substitute), addition of IGFBP7 (100 ng/ml) and VWF did not significantly increase rosetting rates across the VWF concentration range tested (*Figure 7D*). This suggested the need for other serum-derived factors to mediate the IGFBP7 rosetting effect. We suspected that this second cofactor would either be needed in small quantity or present in high abundance in serum, so that even a 2% serum-supplied medium would be adequate to sustain IGFBP7-mediated rosetting. In addition, this co-factor should be able to interact with some of the players identified above (PfEMP1, HS, VWF) to generate IGFBP7-mediated rosettes. One candidate was thrombospondin 1(TSP-1) as it is known to bind to PfEMP1 (*Cooke et al., 1994*; *Baruch et al., 1996*) and VWF (*Pimanda et al., 2004*).

In serum-enriched medium, anti-TSP-1 antibody did not significantly alter baseline rosetting rates. Nevertheless, this antibody significantly blocked the IGFBP7-mediated rosette-stimulation (*Figure 7E*). The specificity of rosette-inhibition by this antibody was validated with experiments using antibody isotype control (*Figure 7—figure supplement 1A*).

When TSP-1 was added to Albumax-supplemented medium, the rosetting rates were lower than those in serum-enriched medium (*Figure 7F*). However, when added together (IGFBP7 + VWF + TSP-1) to the Albumax-supplemented medium, significant rosette-stimulation was observed. A high-level TSP-1 (500 ng/ml) did not induce significantly higher rosetting stimulation than the lower level TSP-1 (10 ng/ml). With the addition of VWF (2 IU/ml) and TSP-1 (10 ng/ml) to Albumax-supplemented medium, IGFBP7 stimulated rosette formation to an extent similar to that of serum-enriched medium. Importantly, without IGFBP7, the presence of TSP-1 and VWF in Albumax-supplemented medium could not increase the rosetting rates, disputing the possibility of this event as a non-specific adhesive effect and reflecting their status as co-factors in IGFBP7-mediated rosetting. Lastly, we quantitated the amount of VWF needed to facilitate IGFBP7-mediated rosetting. When supplemented with IGFBP7 and TSP-1 in Albumax-supplemented medium, VWF as low as 0.125 IU/ml was sufficient to significantly increase rosetting rates, with an optimal increment attained at 0.5 IU/ml

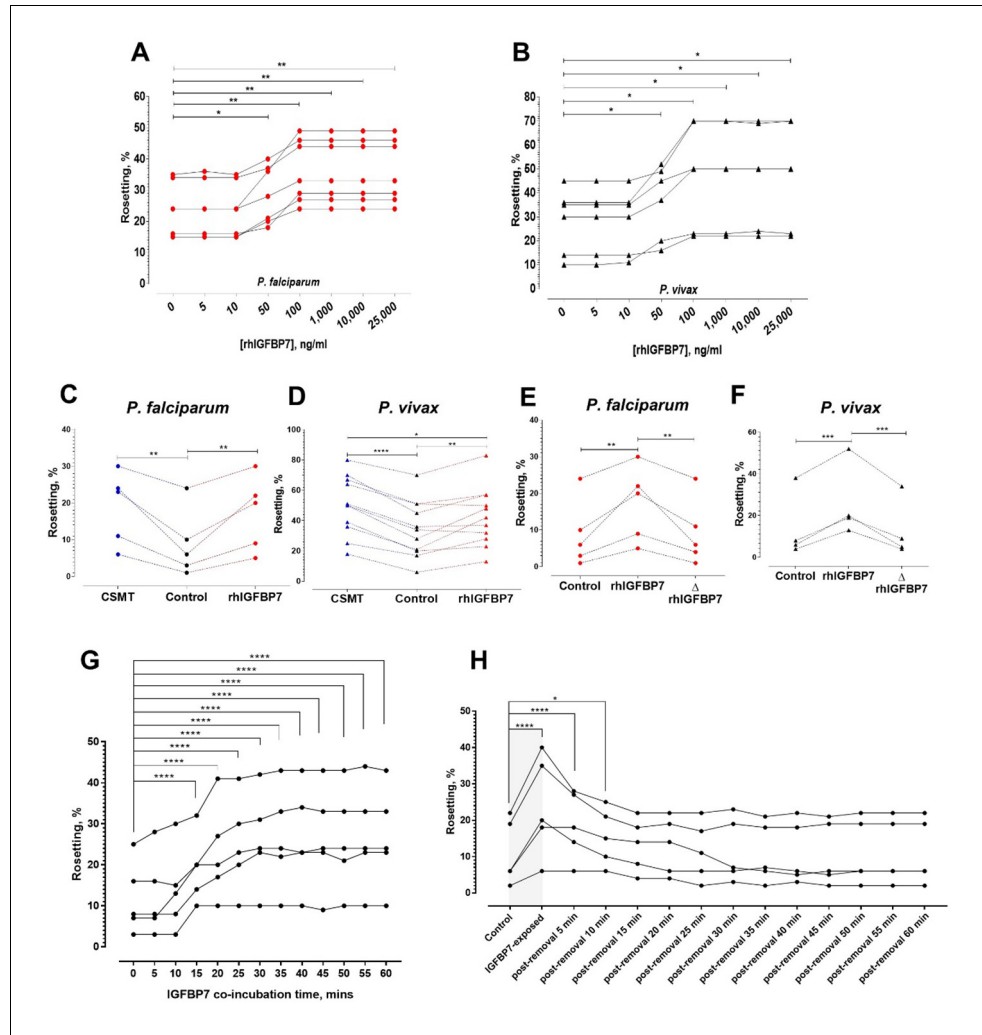

**Figure 4.** Characterization of IGFBP7-mediated rosetting. (A, B) Rosetting rates post-rhIGFBP7-incubation. Significant rosette-stimulation was noted at 50 ng/ml rhIGFBP7 (one-way ANOVA with Dunnett's test: *Pf* (n = 7): p=0.0224; *Pv* (n = 7): p=0.0419) and a plateau was reached after 100 ng/ml (*Pf*: p=0.0045; *Pv*: p=0.0168). (C, D) Comparison of rosette-stimulatory effects between the CSMT and rhIGFBP7 in *P. falciparum* (n = 5) (one-way ANOVA with Tukey's test: control vs. CSMT: p=0.0019; control vs. rhIGFBP7: p=0.0054; CSMT vs. rhIGFBP7: p=0.6866) (C) and *P. vivax* (n = 11) (one-way ANOVA with Tukey's test: control vs. CSMT: p<0.0001; control vs. rhIGFBP7: p=0.0039; CSMT vs. rhIGFBP7: p=0.0108) (D). (E, F) Impact of heat denaturation on rosette-stimulatory effect of rhIGFBP7 in *P. falciparum* (n = 5) (E) (one-way ANOVA with Tukey's test: control vs. rhIGFBP7: p=0.0036; control vs. ΔrhIGFBP7: p=0.9721; rhIGFBP7 vs. ΔrhIGFBP7: p=0.0048) and *P. vivax* (n = 4) (F) (one-way ANOVA with Tukey's test: control vs. rhIGFBP7: p=0.0006; Control vs. ΔrhIGFBP7: p=0.8008; rhIGFBP7 vs. ΔrhIGFBP7: p=0.0004). (G) Changes in lab-adapted *Pf* (n = 5) rosetting rates after incubation with IGFBP7 up till the 60th minute (one-way ANOVA with Dunnett's test: significant increment was detected as fast as the 15th minute (and onwards): p=0.0001). (H) Changes in rosetting rates of lab-adapted *Pf* (n = 5) after IGFBP7-exposure (covered by the gray box), followed by removal of IGFBP7 from the system in the span of 60 min (one-way ANOVA with Dunnett's test: significant rosetting rate increment was seen after IGFBP7 exposure: p=0.0001). The rosetting rates were still significantly higher than the baseline (control) 5 min (p=0.0001) and 10 min (p=0.0265) after IGFBP7-removal. From the 15th to the 60th minute post-IGFBP7 removal, the rosetting rates dropped to levels that were of no significant difference with the baseline values.

The online version of this article includes the following figure supplement(s) for figure 4:

**Figure supplement 1.** IRBC, URBC, and IGFBP7.

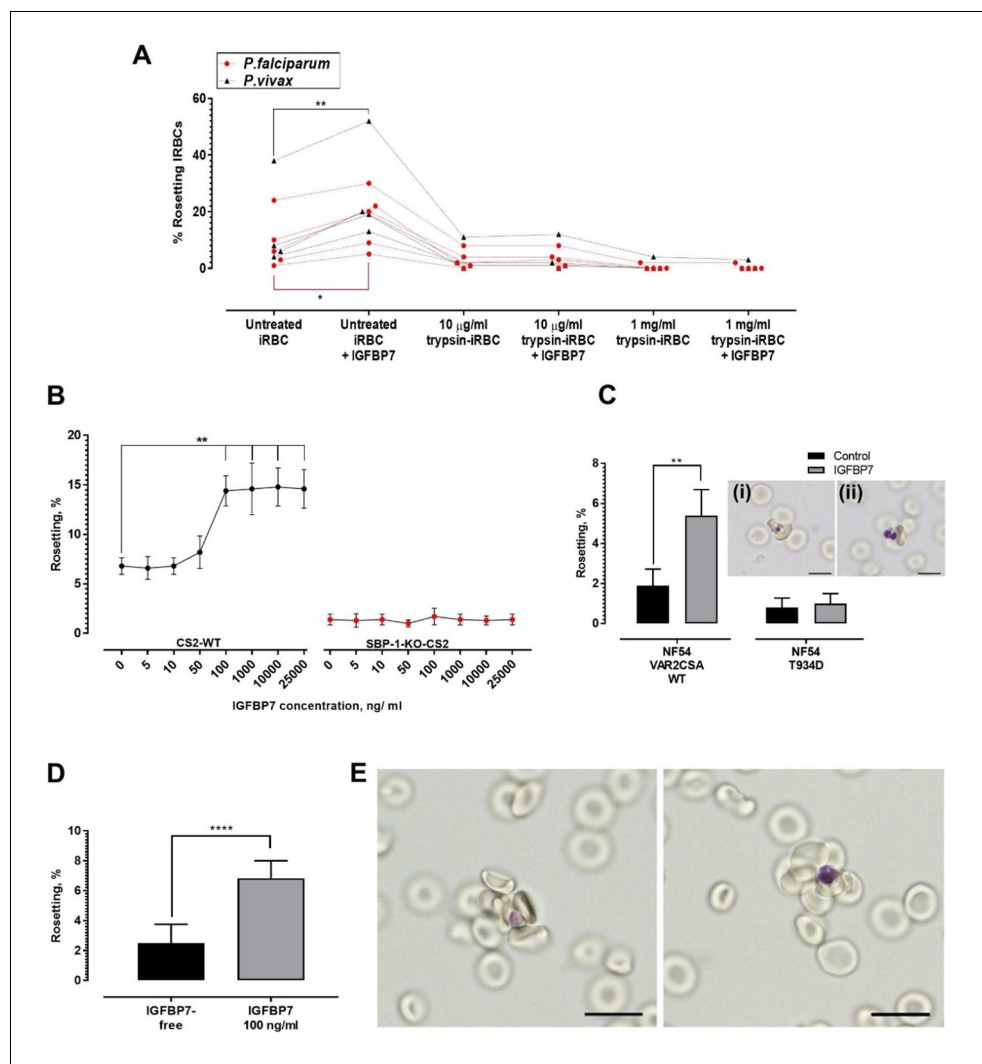

**Figure 5.** Deciphering rosetting ligands involved in IGFBP7-mediated rosetting. (**A**) IRBC-trypsin treatments on IGFBP7-mediated rosetting. Dotted lines demonstrate data collected from same isolates. IGFBP7 increased rosetting in the untreated settings (paired t-test *Pf* (n = 5): p=0.0171; *Pv* (n = 4): p=0.0023) but did not significantly alter rosetting in groups treated with 10 μg/ml trypsin (*Pf*: p=0.3739; *Pv*: p=0.3910) and 1 mg/ml trypsin (*Pf*: same values; *Pv*: p=0.3910). (**B**) Rosetting rates (mean and SD from quintuplicate experiment repeats shown) along rhIGFBP7 concentrations for *Pf*CS2-WT and *Pf*SBP1-KO-CS2. Significant CS2-WT rosette-stimulation was noted from rhIGFBP7 of 100 ng/ml (Friedman with Dunn's test: p=0.0230). No significant changes noted for SBP1-KO-CS2 across the range of IGFBP7 concentrations studied (Friedman with Dunn's test: p>0.9999). (**C**) Effect of IGFBP7 on rosetting rates of NF54_VAR2CSA_WT and NF54_T934D (mean and SD from quintuplicate experiment repeats shown). The Mann-Whitney test was conducted. For NF54_VAR2CSA_WT, IGFBP7 significantly increased the rosetting rates (p=0.0079). For NF54 VAR2CSAT934D, IGFBP7 did not cause any significant change to the rosetting rates of the parasite (p=0.7619). The rosettes formed by NF54_VAR2CSA_WT (**i**) and NF54_VAR2CSA_T934D (**ii**) are small. (**D**) Rosetting rates of the late ring forms (means and SD of nine replicates shown) under IGFBP7-free and IGFBP7-supplied conditions. The Mann-Whitney test was conducted. Rosetting rates were significantly higher in the IGFBP7-supplied group (p<0.0001). (**E**) Rosettes formed by the late ring stage (left) and late trophozoite stage (right) of *P. falciparum*. Scale bar: 10 μm.

The online version of this article includes the following source data for figure 5:

**Source data 1.** Raw data (rosetting rates, %) for the data set presented in the bar graph (*Figure 5C*).
**Source data 2.** Raw data (rosetting rates, %) for the data set presented in the bar graph (*Figure 5D*).

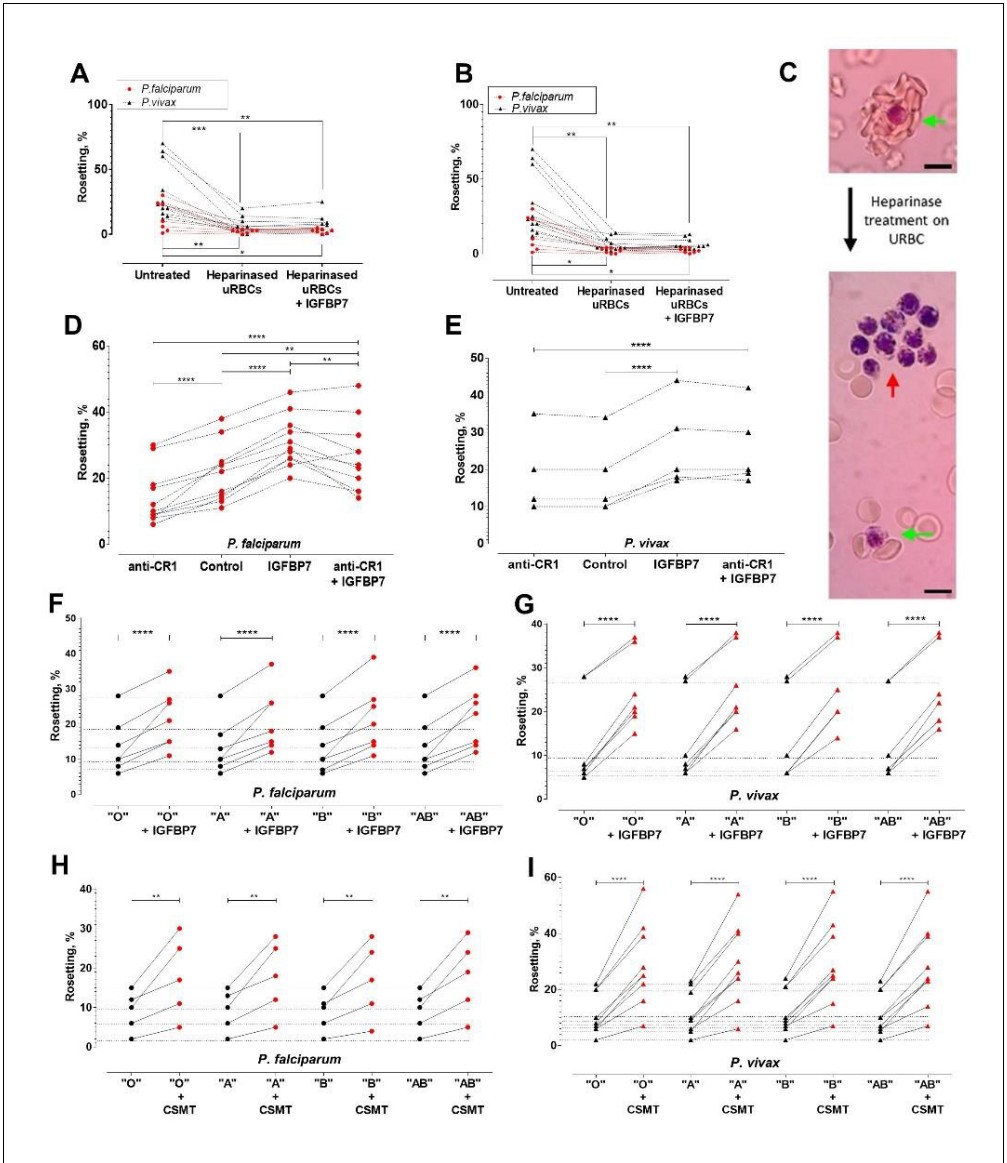

**Figure 6.** Deciphering rosetting receptors involved in IGFBP7-mediated rosetting. (**A**) Heparinase (hep) I treatment (represented as "heparinased" in graph) on URBC hampered IGFBP7-mediated rosetting. Friedman with Dunn's test: *Pf* (n = 7): untreated vs. hep: p=0.0063; hep vs. hep + IGFBP7: p>0.9999; untreated vs. hep + IGFBP7: p=0.0334. *Pv* (n = 11): untreated vs. hep: p=0.0004; hep vs. hep + IGFBP7: p>0.9999; untreated vs. hep + IGFBP7: p=0.0042. (**B**) Hep III treatment (represented as "heparinased" in graph) on U RBC hampered IGFBP7-mediated rosetting. Friedman with Dunn's test *Pf* (n = 7): untreated vs. hep: p=0.0150; hep vs. hep + IGFBP7: p>0.9999; untreated vs. hep + IGFBP7: p=0.0485. *Pv* (n = 11): untreated vs. hep: p=0.0013; hep vs. hep + IGFBP7: p>0.9999; untreated vs. hep + IGFBP7: p=0.0013. (**C**) Rosette size difference between the untreated (left) and heparinase-treated (right) settings of a *P. vivax* isolate. Rosettes are indicated by green arrows, and IRBC-autoagglutination-like clustering is indicated by the red arrow. Pictures taken with Samsung Galaxy W phone camera on Nikon Eclipse E200 light microscope. Scale bars represent 10 μm. (**D**) Anti-CR1 reduced *Pf* (n = 11) rosetting (one-way ANOVA with Tukey's test: p<0.0001). With CR1-blockade, IGFBP7 still managed to induced significant rosette-stimulation (p=0.0057), albeit to a lower degree than Ab-free setting (p=0.0095). (**E**) Anti-CR1 did not inhibit *Pv* (n = 5) rosetting [one-way ANOVA with Tukey's test: p=0.9947]. IGFBP7 increased rosetting (p<0.0001). No significant difference was found between IGFBP7 and anti-CR1+IGFBP7 groups (p=0.9612). Effect of different ABO blood groups on IGFBP7-mediated rosetting for *Pf* (n = 7) (**F**) and *Pv* (n = 7) (**G**). Dotted horizontal lines in these plots matched the same isolates used in each of the different ABO blood group experiments. IGFBP7 significantly increased rosetting rates regardless of the blood groups [one-way ANOVA with Tukey's test: (*Pf*: p<0.0001 for all groups); (*Pv*: p<0.0001 for all groups)]. No significant differences in rosetting rates

*Figure 6 continued on next page*

*Figure 6 continued*

(control and IGFBP7-supplied) across all blood groups for both species. The degree of rosette stimulation by CSMT on *Pf* (n = 5) (H) and *Pv* (n = 9) (I) with URBC of different ABO blood groups. Dotted lines in these plots matched the same isolates used in each of the different blood group experiments. CSMT increased rosetting regardless of the blood groups [one-way ANOVA with Tukey's test: (*Pf*: Group O: p=0.0014; Group A: p=0.0018; Group B: p=0.0033; Group AB: p=0.0010); (*Pv*: p<0.0001 for all blood groups)]. No significant differences were found in rosetting rates (control and CMST-supplied) across all groups for both species.

(*Figure 7G*). In fact, the 20% serum-enriched media that we used for this study contained VWF higher than 0.125 IU/ml (*Figure 7—figure supplement 1B*).

## Quantification of IGFBP7 secretion

*P. falciparum* IRBC-exposed peripheral monocytes secreted significantly more IGFBP7 than their unexposed counterparts or those exposed to URBC (*Figure 8A*). THP-1 also secreted more IGFBP7 after IRBC exposure (*Figure 8B*). Interestingly, parasitemia as low as 0.25% was sufficient to significantly stimulate THP-1 to secrete more IGFBP7. Further increase in parasite density (up to 16% parasitemia) did not significantly increase IGFBP7 secretion any further.

## Knockdown of IGFBP7 expression by THP-1 using an shRNA transduction approach

THP-1 cells were transduced with shRNA lentiviral vectors specific for IGFBBP7 and vectors specific for an unrelated protein, glycophorin C. Non-transduced cells served as wild types (THP-1_WT). Via ELISA, the IGFBP7 production by THP-1 with decreased expression of IGFBP7 (referred as IGFBP7-KD_THP-1) was significantly lower than those of THP-1_WT and THP-1 with decreased expression of glycophorin C (referred to as GlyC-KD_THP-1) after URBC or IRBC stimulation for 18 h (*Figure 8—figure supplement 1*). The URBC- and IRBC- conditioned supernatants from IGFBP7-KD_THP-1 (henceforth referred to as CSKD-U and CSKD-I, respectively) were collected for rosetting assay. CSKD-U and CSKD-I did not significantly increase rosetting rates of the *P. falciparum* parasites (*Figure 8D*), indicating that IGFBP7 is the main rosette-stimulating factor secreted by IRBC-stimulated THP-1. To further validate this, we prepared the culture supernatants (CS) of THP-1_WT, IGFBP7-KD_THP-1 and GlyC-KD_THP-1 in the same the way as we prepared the CSMT and CSUT in earlier experiments where we exposed the cells to IRBC for a longer time (three days) and allowed the cells to grow for a longer time prior to CS harvest. IRBC addition stimulated the THP-1_WT and GlyC-KD_THP-1 to produce a higher level of IGFBP7 than their URBC-exposed counterparts, whereas levels of IGFBP7 remained low and were not significantly different between IRBC and URBC-stimulated IGFBP7-KD_THP-1 (*Figure 8—figure supplement 2A*). Furthermore, addition of anti-IGFBP7 antibody confirmed that IGFBP7 is the major rosette-stimulating factor within the culture supernatant of THP-1_WT and GlyC-KD_THP-1 (*Figure 8—figure supplement 2B*). Importantly, the significant difference between culture supernatant groups 'THP-1_WT_URBC' and 'THP-1_WT_IRBC' (p=0.0018), as well as 'GlyC-KD_THP-1_URBC' and 'GlyC-KD_THP-1_IRBC' (p=0.0013), but not between 'IGFBP7-KD_THP-1_URBC' and 'IGFBP7-KD THP-1_IRBC' (p=0.6011) strongly suggests that IGFBP7 may be one of the key secreted products by the monocytic cells in response to parasite exposure, whereas other factors involved in rosette-stimulation may be secreted at baseline levels with or without the presence of the parasites (*Figure 8—figure supplement 2B*).

## Phagocytosis assessment

We hypothesized that IGFBP7-mediated rosetting could be a strategy used by the parasites to avoid phagocytosis. To test this, we performed a control experiment using Zymosan A (a protein-carbohydrate complex prepared from yeast cell wall, commonly used in phagocytosis assays) and showed that IGFBP7 by itself did not inhibit the phagocytic ability of THP-1 (*Figure 8E*). Unexpectedly, incubation with IGFBP7 increased the phagocytic ability of THP-1. We next tested the phagocytic activity of THP-1 and of human primary monocytes in the presence of IGFBP7-treated culture. As expected, the rosetting rates of the parasite increased after IGFBP7 exposure. (*Figure 8F*). However, IRBC phagocytosis rates by both types of phagocytes were reduced significantly (*Figure 8G*). Subsequently, we repeated this experiment with THP-1 using five different *P. falciparum* lines. All IGFBP7-

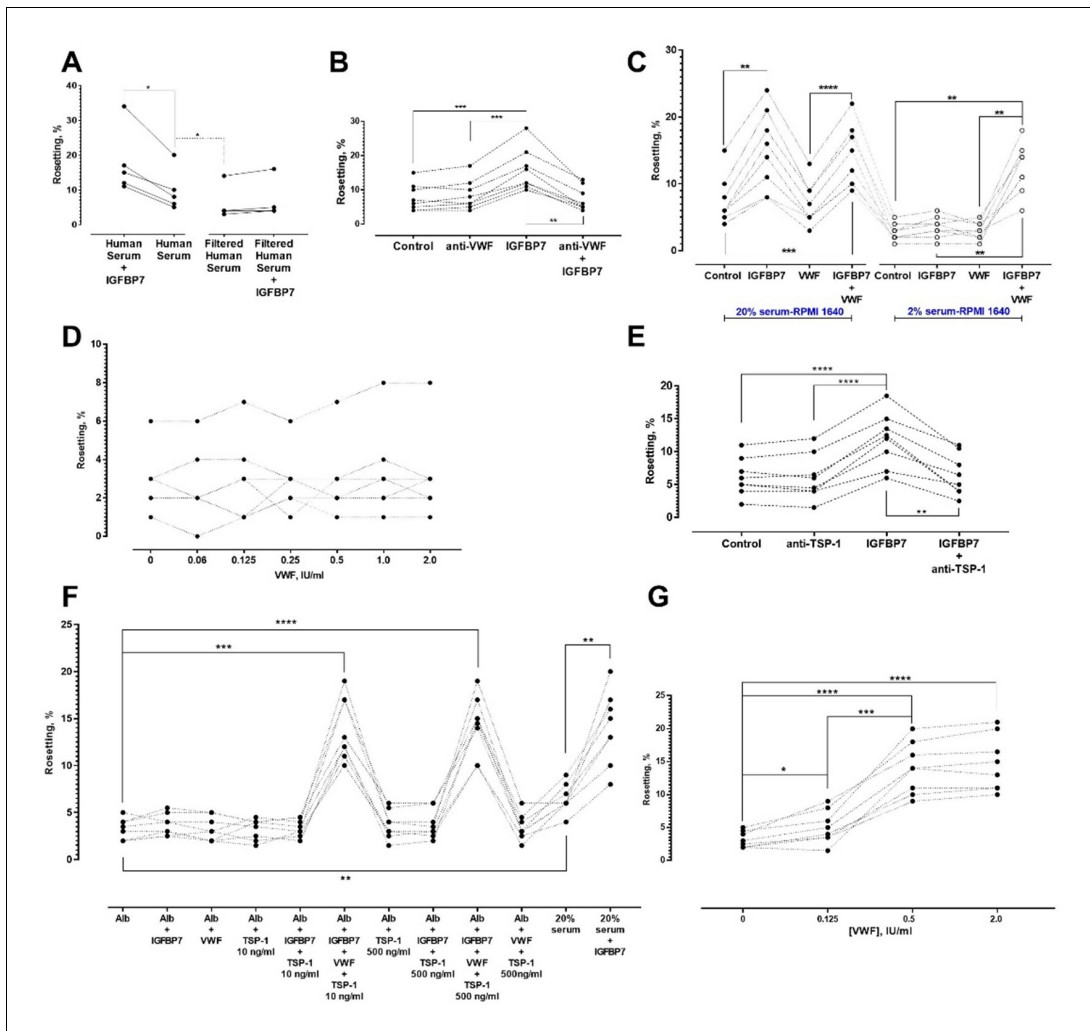

**Figure 7.** Identification of other serum factors involved in IGFBP7-mediated rosetting using lab-adapted *P. falciparum*. (**A**) Rosetting rates in filtered human serum (HS) were lower than those of complete HS (one-way ANOVA with Tukey's test: p=0.0368; n = 5). IGFBP7 stimulated rosetting in HS-supplied environment (p=0.0276). IGFBP7 did not alter rosetting in filtered HS (p=0.0664). (**B**) IGFBP7 stimulated rosetting (p=0.0002; n = 8), whereas anti-VWF IgG did not significantly alter rosetting (p=0.3039). No significant changes were found in 'anti-VWF IgG +IGFBP7' (p=0.9096). (**C**) In 20% HS-enriched medium (20% HSM), 'IGFBP7 and IGFBP7+VWF' showed higher rosetting than the control (one-way ANOVA with Tukey's test: p=0.003 for both comparisons; n = 8). VWF did not alter rosetting (p=0.8652). No significant difference was found between 'IGFBP7' and 'IGFBP7+VWF' (p=0.7853). Rosetting in 'IGFBP7' was higher than in 'VWF' (p<0.0001). In 2% HSM, 'IGFBP7' (p=0.1832) and 'VWF' (p=0.9876) did not significantly alter rosetting. Rosetting in 'IGFBP7+VWF' was increased (p=0.0014), and was higher than 'IGFBP7' and 'VWF' (p=0.0030 for both comparisons). No significant difference was found between 'VWF+IGFBP7' from both serum settings (p=0.2861). (**D**) Rosetting (n = 7) in 0.25% Albumax-enriched medium (Alb) supplied with 100 ng/ml IGFBP7 and different concentrations of VWF. No significant difference was found across the VWF concentrations tested (one-way ANOVA with Dunnett's test: p>0.3 for all comparisons with 'VWF-free'). (**E**) In 20% HSM, there was no significant difference between control and 'anti-TSP-1' (one-way ANOVA with Tukey's test: p=0.9961, n = 8). A significant difference was recorded between control and 'IGFBP7' (p<0.0001), but not between control and 'anti-TSP-1 + IGFBP7' (p=0.9125). A significant difference was found in 'anti-TSP-1' vs. 'IGFBP7' (p<0.0001), and 'IGFBP7' vs. 'IGFBP7 + anti-TSP-1' (p=0.0022). (**F**) Rosetting (n = 8) in Alb was lower than in 20% HSM (p=0.0047). In HSM, IGFBP7 increased rosetting (p=0.0097). No significant changes were seen in comparisons of Alb-control with: IGFBP7 (p=0.7499), VWF (p>0.9999), TSP-1 10 ng/ml (p>0.9999), TSP-1 500 ng/ml (p=0.9491), IGFBP7 + TSP-1 10 ng/ml (p>0.9999), and IGFBP7 + TSP-1 500 ng/ml (p=0.9341). Rosette-stimulation was noted in 'IGFBP7 + VWF + TSP-1 10 ng/ml' (p=0.0002) and 'IGFBP7 + VWF + TSP-1 500 ng/ml' (p<0.0001). No significant difference was noted between 'IGFBP7 + VWF + TSP-1 10 ng/ml' and 'IGFBP7 + VWF + TSP-1 500 ng/ml' (p=0.9998), and 'IGFBP7 + VWF + TSP-1 10 ng/ml' vs. '20%HSM + IGFBP7' (p>0.9999). 'IGFBP7 + VWF + TSP-1

*Figure 7 continued on next page*

*Figure 7 continued*

500 ng/ml' was not significantly different from 'HSM + IGFBP7' (p=0.9997). (G) In 0.25% Alb supplied with 100 ng/ml IGFBP7, 10 ng/ml TSP-1 and 0.125 IU/ml VWF, rosette-stimulation was noted, as compared to a VWF-free group (one-way ANOVA with Tukey's test: p=0.0421, n = 8). Rosetting increased with VWF concentrations (p<0.0001 for 0.5 and 2.0 IU/ml compared to VWF-free). No significant difference was seen in rosetting rates between VWF of 0.5 IU/ml and 2.0 IU/ml (p=0.2126).

The online version of this article includes the following figure supplement(s) for figure 7:

**Figure supplement 1.** Control experiments.

---

incubated *P. falciparum* lines prior to THP-1 exposure formed more rosettes (*Figure 8H*). They were significantly less phagocytosed than their non-IGFBP7-exposed counterparts (*Figure 8I*). Individual phagocytes could engulf non-rosetting IRBC (*Figure 8J*); however, successful engulfment of a rosette was only observed when several phagocytes were recruited (*Figure 8K and L*).

## Discussion

Rosetting is a common characteristic of late stage-IRBC in human malaria parasites, occurring frequently in *P. falciparum* and *P. vivax* (*Lee et al., 2014*). It has been proposed to provide a survival advantage for the parasites (*Moll et al., 2015*). Earlier studies have shown that rosetting occurs between the direct interactions of the parasite-derived ligands on the IRBC (i.e. PfEMP1, RIFIN, and STEVOR proteins for *P. falciparum*) with various receptors on the URBC (*Lee et al., 2014*; *Niang et al., 2014*; *Rowe et al., 1997*; *Barragan et al., 2000*; *Cserti and Dzik, 2007*; *Chen et al., 1998*; *Goel et al., 2015*).

Here we demonstrated the existence of a different type of rosetting, which we have called 'type II rosetting' that does not result from the direct interaction of IRBC with URBC. It was observed in all the *P. falciparum* and *P. vivax* isolates tested. This type II rosetting differs from the classical type I rosetting as it requires bridging by soluble mediators: IGFBP7, VWF, and TSP-1 between a rosetting ligand on IRBC and HS expressed by URBC. The fast rosette-stimulating effect by the protein and its fast reversion after the protein removal from the culture suggest that IGFBP7 does not mediate rosetting via irreversible binding to neither the rosetting receptor nor ligand. Instead, it is more likely to be mediated by weaker forces. It also suggests that these soluble mediators need to be present at a minimum concentration for the rosettes to occur.

We have shown that PfEMP1 is likely the principal *P. falciparum* rosetting ligand in this IGFBP7-mediated type II rosetting via usage of trypsin treatments, genetically modified *P. falciparum* clones that cannot surface-express PfEMP1, and the late ring stages of *P. falciparum*, the stage of maturation that manages to surface-express only one rosetting ligand, PfEMP1. However, we cannot fully dismiss the involvement of other rosetting ligands such as STEVOR or RIFIN as these proteins are encoded by multigene families and there is a lack of tools to assess and evaluate the implication of these proteins thoroughly. To our surprise, *P. vivax* IRBC also interacted with IGFBP7 similarly. There are no PfEMP1 orthologues in *P. vivax*. Based on the rosetting trend of *P. vivax* post-trypsin treatments, we postulate that *P. vivax* has multiple rosetting ligands with different trypsin sensitivities, and the one required by IGFBP7 is highly sensitive to trypsin. IGFBP7 requires the HS moieties on URBC to exert their rosette-stimulatory effect. Interestingly, removal of HS from the surface of URBC by heparinase caused the clumping of untreated IRBC (which harboured both rosetting ligands and receptors) when supplied with IGFBP7. Importantly, without enzymatic interference, the presence of this protein does not induce non-specific binding of URBC to each other, or autoagglutination-like clumping of IRBC. We hypothesized that the IGFBP7-mediated binding occurs preferably between the URBC and IRBC under normal circumstances possibly as a result of electrostatic differences between the URBC and IRBC (*Tokumasu et al., 2012*).

IGFBP7 requires other serum factors, namely VWF and TSP-1 to exert its rosette-stimulating effect. For healthy individuals, the VWF levels in the serum range from 0.48 to 1.24 IU/ml (median 0.84 IU/ml), whereas individuals with underlying pathological conditions have much higher levels of serum VWF (*Terpos et al., 2013*; *Kastritis et al., 2016*). Serum TSP-1 levels in healthy individuals vary greatly (0–12060 ng/ml) (*Liu et al., 2015a*; *Rouanne et al., 2016*). We found that under serum-free conditions (Albumax- supplemented medium), concentrations of VWF as low as 0.5 IU/ml and of

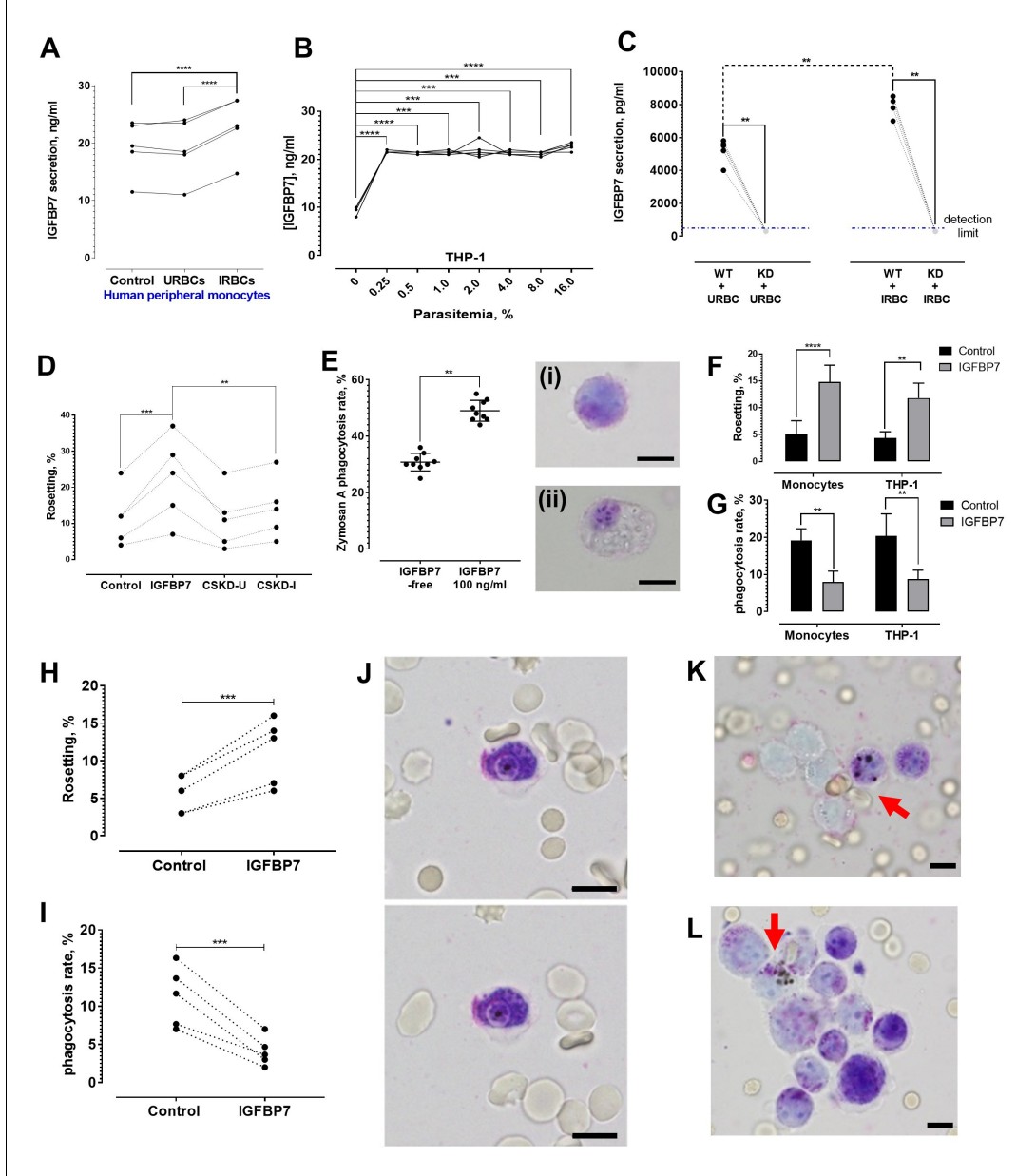

**Figure 8.** IGFBP7 secretion upon parasite exposure and phagocytosis assessment. (**A**) Parasite (lab-adapted *P. falciparum* strain 3D7) exposure increased IGFBP7 secretions by human peripheral monocytes (from five donors) [paired t-test control vs. IRBC: p<0.0001; URBC vs. IRBC: p<0.0001]. (**B**) Secretion of IGFBP7 by THP-1 upon exposure to different parasite lines (n = 5) at different parasite density. Parasite exposure yielded significantly higher readings than the parasite-free control (one-way ANOVA with Dunnett's test: p=0.0001). (**C**) IGFBP7 secretion by WT_THP-1 (WT) and IGFBP7-KD_THP1 (KD) exposed to URBC and IRBC for 18 h. The blue dotted line shows the detection limit (400 pg/ml). For control groups (incubated with URBC), WT showed significantly higher IGFBP7 secretion than the KD (Mann-Whitney test: p=0.0079, n = 5). For the IRBC-exposed groups, WT also showed significantly higher IGFBP7 secretion than the KD (p=0.0079, n = 5). IGFBP7 secretion by WT post-IRBC exposure was significantly higher than its control (p=0.0079, n = 5). No significant difference was found in IGFBP7 secretion in the KD between the IRBC-exposed and control settings (p>0.9999, n = 5). (**D**) A significant difference in rosetting (n = 5) was found: control vs. IGFBP7 (one-way ANOVA with Tukey's test: p=0.0001), and IGFBP7 vs. CSKD-I (p=0.0014). No significant difference was found: control vs. CSKD-U (p=0.9944), control vs. CSKD-I (p=0.4118), and CSKD-U vs. CSKD-I (p=0.2977). (**E**) A control experiment (n = 9) to assess effect of IGFBP7 on phagocytosis ability of THP-1 using zymosan A, with insets showing THP-1 before (**i**) and after (**ii**) engulfing zymosan A, in Giemsa-wet mount, 1000× magnification; scale bars: 10 μm. IGFBP7 enhanced phagocytosis of

*Figure 8 continued on next page*

*Figure 8 continued*

THP-1 (Wilcoxon matched pair-signed rank test p=0.0039). (**F**) Rosetting rates of *Pf* co-incubated with monocytes and THP-1 under IGFBP7-free and IGFBP7-supplied conditions (means and SD of quintuplicates shown). From two-way ANOVA with Sidak's multiple comparison test, IGFBP7 increased rosetting rates (p<0.0001 and p=0.0013 for monocyte and THP-1, respectively). No significant difference was found between the controls (IGFBP7-free) (p=0.9968), as well as between the IGFBP7-supplied conditions of the two groups (p=0.3465). (**G**) IRBC-phagocytosis rates of monocytes and THP-1 under IGFBP7-free and IGFBP7-supplied conditions (means and SD of quintuplicates shown). From two-way ANOVA with Sidak's multiple comparison test, phagocytosis rates were lower in the IGFBP7-supplied group (p=0.0015 and 0.0011 for monocyte and THP1, respectively). No significant difference was found between the controls (IGFBP7-free) (p=0.9595), as well as between the IGFBP7-supplied conditions of the two groups (p=0.9873). (**H, I**) IGFBP7-exposure affects laboratory-adapted *pf* IRBC rosetting (n = 5) (paired t-test p=0.0001, t = 14.88, df = 4) (**H**) and phagocytosis (paired t-test p=0.0009, t = 8.98, df = 4) (**I**). (**J**) Engulfment of a non-rosette-forming IRBC by a peripheral monocyte. Pictures taken 10 s apart. (**K, L**) Phagocytosis (arrow) of rosettes, in Giemsa-wet mount, 1000× magnification, using an Olympus BX43 light microscope with built-in camera; scale bars: 10 μm.

The online version of this article includes the following source data and figure supplement(s) for figure 8:

**Source data 1.** Raw data (rosetting rates, %) for the data set presented in bar graph (8F).
**Source data 2.** Raw data (phagocytosis rates, %) for the data set presented in bar graph (8G).
**Figure supplement 1.** IGFBP7 secretion by THP-1 of different transduction status.
**Figure supplement 2.** THP-1-derived IGFBP7 and its effect on rosette-stimulation.

TSP-1 at 10 ng/ml were enough to optimally facilitate IGFBP7-mediated rosette-stimulation at IGFBP7 of 100 ng/ml (the minimum concentration needed to optimally stimulate type II rosetting). The rosette-stimulation by the presence of these three proteins was comparable to that of 20% serum-enriched medium supplied with IGFBP7, reinforcing that IGFBP7 is the limiting factor for the rosette-stimulation.

Based on the data presented here, we propose the following mechanism of interactions for type II rosetting (*Figure 9*). IGFBP7 binds to HS on URBC; the interaction between IGFBP7 and cell surface HS has been demonstrated and well-characterized (*Sato et al., 1999*). IGFBP7 has also been shown to bind to the D4-CK region of VWF (*van Breevoort et al., 2012*; *Lenting et al., 2012*; *Bryckaert et al., 2015*). Although heparin is found to interact with the A1 region of VWF (*Bryckaert et al., 2015*), it is likely that HS does not interact directly with VWF (*Denis et al., 1993*) as we did not observe any rosetting when VWF was added alone in the absence of IGFBP7. Therefore, the HS on URBC interacts with IGFBP7, which also interacts with VWF. In turn, VWF interacts with PfEMP1 on IRBC via TSP-1 (*Cooke et al., 1994*; *Bryckaert et al., 2015*). While the extracellular domain of PfEMP1 that binds to TSP-1 has yet to be identified, it should be noted that TSP1 has been commonly associated with PfEMP (*Cooke et al., 1994*; *Janes et al., 2011*). The pervasive role of TSP-1 may explain why a wide range of clinical isolates and laboratory-adapted parasite lines were capable of responding positively to IGFBP7 addition.

Induction of type II rosetting in *P. falciparum* and *P. vivax* is not attributed solely to IGFBP7. Previously, it has been shown that CFD in the serum can stimulate rosetting (*Luginbühl et al., 2007*). This molecule was also identified in our proteomic study. Experiments with anti-CFD antibodies showed that CFD could also induce type II rosetting, but to a lesser extent. The knock down of IGFBP7 expression in THP-1 demonstrated that IGFBP7 is a major monocyte-derived rosette-stimulating factor. Culture supernatant from IGFBP7-KD_THP-1 collected after 18 h of parasite exposure could not induce rosette-stimulation. Other rosette-stimulating factors may be secreted by the cells much later. Interestingly, IGFBP7 and VWF are components in Weibel-Palade bodies, the storage granules of endothelial cells (*van Breevoort et al., 2012*). Future work should characterize the effect of IRBC on the secretion of IGFBP7 by endothelial cells. Of note, the reported physiologic and pathological serum concentrations of IGFBP7 vary greatly, where most of the reported normal serum IGFBP7 concentrations fall below 50 ng/ml, but the serum IGFBP7 levels in different pathological conditions (e.g. various vascular disorders and cancers) are higher (as high as ~1000 ng/ml) (*Kutsukake et al., 2008*; *Shersher et al., 2011*; *Liu et al., 2015b*; *Shaver et al., 2016*; *Barroso et al., 2016*). Therefore, the working concentration of IGFBP7 in this study (100 ng/ml) is still within the pathophysiologic range in clinical settings. It would be interesting to compare the serum/plasma IGFBP7 levels

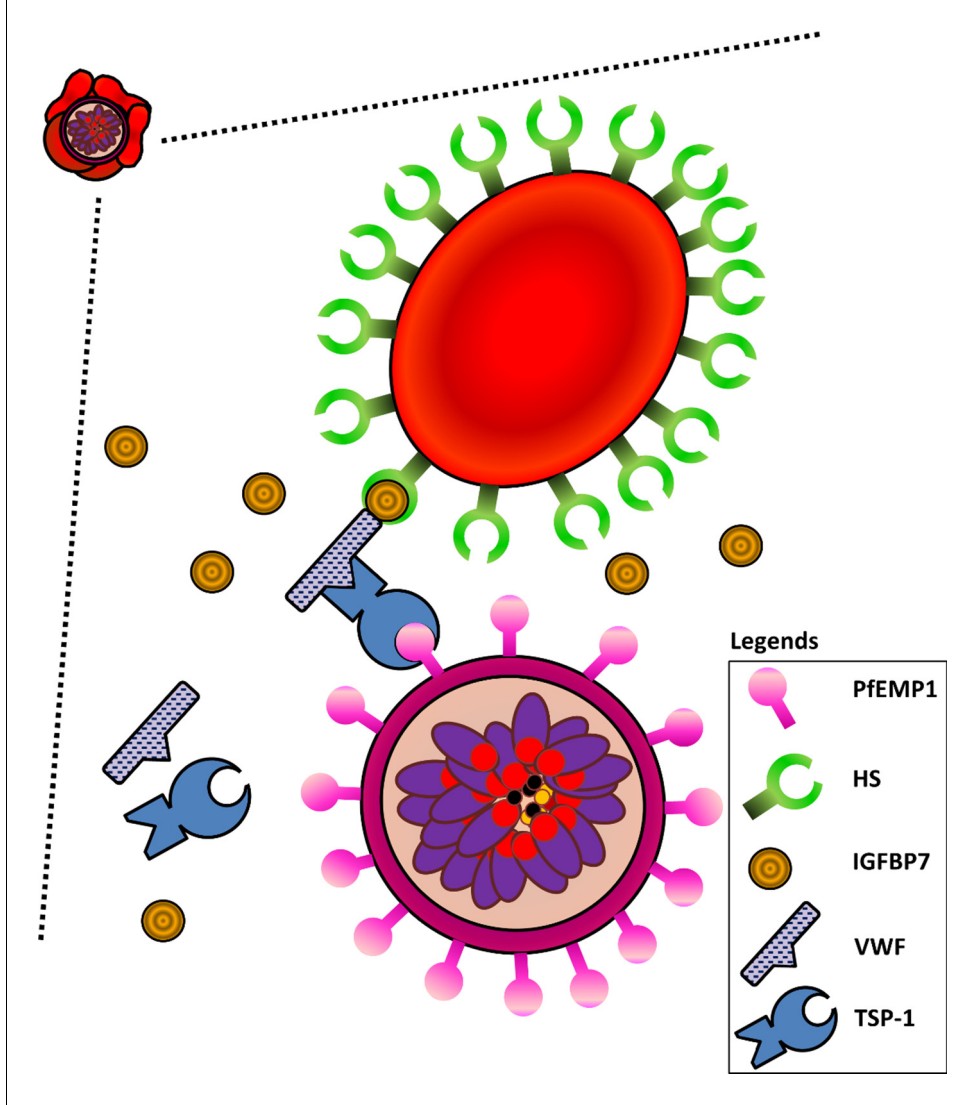

**Figure 9.** Schematic diagram illustrating interactions of HS on URBCs, IGFBP7, VWF, TSP-1, and PfEMP1 on IRBCs. IGFBP7 binds to HS on the surface of URBCs. IGFBP7 also interacts with VWF. VWF can interact with PfEMP1 on the IRBC via TSP1. Without sufficient amounts of all components involved in the system, this IGFBP7-mediated rosetting will not occur.

of uncomplicated and severe malaria patients from the same area to understand better the role of IGFBP7 in malaria pathogenesis.

The importance of monocytes/macrophages in eliminating *Plasmodium* during its course of infection has been reported (*Groux and Gysin, 1990*; *Theander, 1992*). Peripheral monocytes (*Turrini et al., 1992*; *Muniz-Junqueira and Tosta, 2009*), as well as the tissue-resident macrophages (*Tosta et al., 1983*; *Sponaas et al., 2009*), have been shown to engulf IRBC readily. To survive, parasites must counter or avoid this host's immune responses. The ability to perceive a phagocyte's secreted IGFBP7 as a signal of 'approaching threats' and to respond by rosetting may provide survival advantage to the parasites.

In conclusion, the host-derived IGFBP7 is used as an 'incoming phagocyte signal' by the IRBC, and the IRBC in turn use this protein, along with two serum factors, VWF and TSP-1, to stimulate rosette formation, which acts as an immune-evasion strategy by hampering phagocytosis of the IRBC. It is hoped that future clinical studies will investigate associations between IGFBP7 and malaria pathogenesis and immunity.

## Materials and methods

Experiments were performed with isolates of late-stage *P. falciparum* and *P. vivax* late stages unless stated otherwise (materials used in the experiments and the work flow are available in *Supplementary files 1*, *4–6*). All incubations were conducted for 1 h under in vitro cultivation conditions unless stated otherwise. Experiments with only one cell line or one parasite line were performed with biological replicates, where the experiments were conducted with multiple sets/batches of the cell/parasite cultures under the same cultivation conditions.

### Ethical statement

Malaria-infected samples were collected in Shoklo Malaria Research Unit (SMRU) under approved ethics: OXTREC 04–10 (University of Oxford, UK); TMEC 09–082 (Ethics Committee, Faculty of Tropical Medicine, Mahidol University, Thailand).

### Cell lines

Human monocytic THP-1 cell line (Source: ATCC) was used in this study. The cells were tested Mycoplasma-free using the MycoAlert Plus Mycoplasma Detection Kit (Lonza).

### Blood sample processing

Clinical isolates (uncomplicated malaria cases) from SMRU were recruited. Blood (volume: 3 ml) was collected using a BD Vacutainer with lithium heparin anticoagulant. Blood groups were determined with TransClone Anti-A and Anti-B antibodies. Blood samples were centrifuged at 1500 g for 5 min. Plasma was removed, and the buffy coat was carefully collected. A CF11-packed column was used to filter the remaining leukocytes. The parasites were matured in vitro under culture conditions of 5% haematocrit using 20% human homologous serum-enriched RPMI 1640 medium for *P. falciparum*, and McCoy's 5A for *P. vivax*, under gas conditions of 4% $CO_2$ and 3% $O_2$.

### Human leukocytes-*Plasmodium* rosetting correlation testing

Leukocytes and red blood cells (RBC) from clinical samples were isolated and divided into two groups. One group consisted of only RBC. In another group, RBC and leukocytes at a physiologic ratio of 500:1 were matured in vitro, prior to rosetting assay (*Lee et al., 2014*; *Lee et al., 2013*; *Chotivanich et al., 1998*). There are different wet mount-based techniques that can be used for rosetting assay (*Lee et al., 2014*; *Lee et al., 2013*; *Chotivanich et al., 1998*; *Udomsangpetch et al., 1992*; *Udomsangpetch et al., 1989*; *Treutiger et al., 1998*; *Ribacke et al., 2013*; *Adams et al., 2014*). We compared three commonly used techniques and validated that they can be used interchangeably (*Supplementary file 7*), as elaborated in the following sections.

### Rosetting assay

Rosetting assay was conducted when 70% of the parasite population reached late stages (late trophozoites and schizonts) unless stated otherwise. The parasite culture suspension was stained subvitally with Giemsa (5% stain working concentration) for 20 min. Subsequently, 7.6 µl stained suspension was pipetted onto a clean glass slide, immediately covered with a 22 × 32 mm glass cover slip. The wet mount was examined with light microscope using 1000× magnification. Rosetting rate (percentage of rosetting IRBCs) was defined as the percentage of IRBC (over 200 recruited IRBC) that form rosettes. Experiments were conducted in a blinded manner.

### Comparison of different rosetting assay methods

There are different microscopy-based techniques that can be used for rosetting assay, namely the unstained, Giemsa-stained, and the fluorescent dye acridine orange (AO)-stained wet mounts, but none of them have been thoroughly validated and compared. When the majority of the *P. falciparum* culture population (laboratory-adapted lines 3D7, CS2-WT, FVT201, MKK183, WPP3065) reached late stages, the culture suspension was used for this experiment. For each parasite line, the culture suspension was divided into three parts. One aliquot was stained with 5% Giemsa subvitally for 15 min prior to wet mount preparation for rosetting assay. Another was stained with acridine orange (working concentration 2 µg/ml) for 15 min prior to wet mount for rosetting assay. The third

aliquot was used for unstained wet mount preparation prior to rosetting assay. Rosetting rates were determined and compared.

The late stage-IRBCs were purified with MACS-LD columns. Only the yields with IRBC purity of at least 90% were used. The purified IRBCs were divided into two groups. One was mixed with URBCs to make a 3% parasitemia packed cell mixture. The other group was used as a purified IRBC group. Packed URBCs from two healthy individuals were used as controls. Each of these groups was further divided into two parts, where one was exposed to IGFBP7 (100 ng/ml) and the other acted as an IGFBP7-free control. All groups were suspended with culture medium, and incubated for 1 h at in vitro cultivation conditions prior to rosetting assay using the Giemsa-wet mount, acridine orange-wet mount, and unstained wet mount methods described in the previous paragraph.

In our hands, these three techniques yielded comparable rosetting rates with or without IGFBP7 (*Supplementary file 7*). Importantly, in all techniques applied, IGFBP7 did not exert a clumping effect on groups consisting of only URBC and IRBC (*Figure 4—figure supplement 1*), demonstrating the IRBC-URBC specificity of the IGFBP7 interaction. Of note, the formation of some IRBC aggregates when enriched IRBC are cultured has been described previously by *Adams et al. (2014)*. However, addition of IGFBP7 did not aggravate the IRBC aggregate formation or enlarge the size of the aggregates (data not sown). Thus, we have validated that the currently available microscopy-based rosetting assays have similar reliability and can be used interchangeably.

## Human monocytes and neutrophils on rosetting

Three Percoll gradients were prepared from the isotonic Percoll (9 parts Percoll stock + 1 part 10× PBS), namely 81% Percoll, 68% Percoll, and 55% Percoll. A 15 ml conical centrifuge tube was layered with 3 ml of 81% Percoll solution, followed by 3 ml of 68% Percoll solution. Blood from healthy donors was centrifuged for 5 min at 1500 g. Subsequently, three-quarters of the plasma supernatant were removed, followed by careful collection of the leukocyte-rich buffy coat layer. Uptake of RBC must be avoided as much as possible. The leukocytes were suspended with 3 ml of 55% Percoll, and carefully transferred onto the Percoll layered column prepared. The gradient tube was centrifuged at 1500 g for 20 min. The upper two-thirds of the top layer was removed. The remaining one-third (the '55–68' interface zone, i.e. monocyte-rich PBMCs) was collected. After that, the upper two-thirds of the second layer was removed, and the '68–81' interface (neutrophil-rich PMNs) was collected in another conical centrifuge tube. The separated monocytes and neutrophils were washed with RPMI 1640 medium. After that, trypan blue exclusion examination and Giemsa-stained blood smear examination were performed to evaluate viability, cell numbers, and purity of cell population harvested. The cells were suspended in RPMI 1640 and incubated in petri dishes for 3 h at 37°C. For the monocyte group, cells that adhered to the petri dish (monocytes) were retained for experiments whereas the non-adhered cells were removed from the petri dishes.

*P. falciparum* (three laboratory-adapted lines: 3D7, FVT402, FVT201, and one clinical isolate RDM00036) were cultured (5% haematocrit, 3% parasitemia). Culture suspensions were incubated with or without neutrophils (RBC: neutrophil ratio 1000: 1) or monocyte (RBC: monocyte ratio 10,000: 1) from individual donors. Rosetting assay was performed afterwards. In a separate experiment, monocytes from two healthy donors were purified via Ficoll density gradient concentration method, followed by sorting of CD14$^+$ microbeads. The impacts of purified CD14$^+$ monocytes and CD14$^-$ peripheral blood mononuclear cells (PBMC), as well as the human monocytic THP-1 cell line on rosetting rates of *P. falciparum* lines were assessed using laboratory-adapted *P. falciparum* lines (FVT402, 3D7, and MKK183). For each parasite line, three batches of cultures (thawed from vials that were cryopreserved at different times) were used for three experiment replicates.

## THP-1 and *P. falciparum* rosetting

Human monocytic THP-1 cell line (*Mycoplasma*-free) was cultured [10% FBS -enriched RPMI 1640] and expanded with two methods; one being cultivated with stringent control of cell density below $10^6$ cells/ ml as undifferentiated THP-1 (UT) cells. The other was allowed to replicate until the cell population reached $6 \times 10^6$ cells/ ml for differentiation into macrophage-like THP-1 (MT) cells. Supernatant (2 ml) collected from *P. falciparum* culture (rich with parasite antigens) was added to the second group, along with interferon gamma (IFNγ) (final concentration 50 ng/ml). Three days later, the culture medium was discarded, then $2.5 \times 10^5$ cells were transferred into each well of a

48-well flat bottom culture plate, whereas the remaining cells continued to be cultured in the flask. All cell cultures were replenished with fresh culture medium (5% FBS-enriched RPMI 1640) without addition of *P. falciparum* culture supernatant and IFNγ. Two days later, cell counts were performed. The culture supernatant (CS) from the 48-well plate (1 ml for each well) was collected as CSMT for subsequent experiments. The preparation was repeated with the UT cells to collect CSUT. The attached, differentiated MT cells in the culture flask were harvested by removing the culture medium, followed by addition of pre-chilled 1× PBS into the culture flask and incubation on ice for 10 min. UT and MT of different cell numbers were tested in the rosetting assay in triplicate for each parasite line. CSMT and CSUT (used in dilution equivalent to that of $1 \times 10^6$ cells), and IFNγ (50 ng/ml) were also tested in the rosetting assay.

## Identification of mediators mediating rosette-stimulation

CSMT was fractionated into lipophilic (lipid) and aqueous (aq) compartments using Folch's chloroform-methanol extraction method (*Folch et al., 1957*). Briefly, chloroform-methanol extraction mixture (2:1 ratio) was prepared. Washing liquid consisted of chloroform, methanol, and distilled water in a ratio of 3:48:47 was prepared. CSMT (1 ml, from culture with cell density of $10^6$ cells/ml) was mixed with 19 ml extraction mixture. The mixed liquid was washed with 4 ml of distilled water and allowed to settle for a few minutes. Two phases of liquid formed from this. The upper portion (around 40% of the total volume) being the aq fraction, and the lower part being the lipid fraction. The aq fraction was collected separately. The remaining liquid was washed gently with washing mixture three times to remove the interphase. After that, the lipid phase was collected. The collected aq and lipid fractions were dried with a vacuum concentrator. After that, the pellet was suspended with 500 μl distilled water. Vortexing was applied to facilitate solubilization of the lipid pellet. These fractions were tested with rosetting assay. The aq fraction was further subjected to size-based fractionation using Vivaspin20 twin PES membrane (30,000 MWCO) concentrator and tested with rosetting assay. A separate experiment was conducted with laboratory-adapted *P. falciparum* lines (3D7, FVT402, FVT201, MKK183) to compare the rosette-stimulating effect of the aq ≤30 kDa fraction and the aq ≤30 kDa fraction that was heated for 1 h at 56°C. Subsequently, the aq fraction (≤30 kDa) was digested for mass spectrometry analysis using an Orbitrap Fusion mass spectrometer.

## Mass spectrometry

Samples were in-solution digested. Initial denaturation was done with 8M urea in 50 mM Tris-HCl pH 8.5. Following denaturation, proteins were reduced in 25 mM Tris-(2-carboxyethyl) phosphine (TCEP), alkylated with 55 mM chloroacetamide (CAA), and further diluted with 100 mM triethylammonium bicarbonate (TEAB) to achieve <1M urea concentration. Two-step enzyme digestion with lysyl endopeptidase (LysC) and trypsin was performed for 4 h (1:100 –enzyme/protein ratio) and 18 h (1:100), respectively. After acidification with 1% trifluoroacetic acid (TFA), desalting was done using Sep-Pak C-18 columns. The organic phase was evaporated in the vacuum centrifuge. For high pH reverse phase initial separation sample was re-suspended in 10 mM ammonium formate/5% acetonitrile. Two-hundred minutes continuous gradient separation (Solvent A: 10 mM ammonium formate pH10.5/Solvent B: 10 mM ammonium formate pH10.5/90% acetonitrile) was performed on an ÄKTA Micro system using Gemini 5 u/C-18/110A, 150 mm × 1 mm column. Collected fractions were combined into 14 fractions, evaporated and used for mass spectrometry analysis. Mass spectrometry analysis was performed on an Orbitrap Fusion mass spectrometer coupled to nano-ultra-high-performance liquid chromatography (UHPLC) Easy nano liquid chromatography (nLC 1000 system). Fractions were injected and separated on in-house prepared (C-18 ReproSil Pur Basic beads 2.5 um) fused silica emitter column 20 cm × 75 μm in 75 min gradient (solvent A: 0.1% formic acid; solvent B: 0.1% formic acid/99.9% acetonitrile) in data dependent mode using Orbitrap (OT) and Ion trap (IT) detectors simultaneously (speed mode −3 s cycle) with ion targets and resolution (OT-MS 2xE5, resolution 60K, OT-MS/MS 3.5E4, resolution 15 k; IT-MS/MS 2E4, Normal scan). Peak lists were generated with Proteome Discoverer 1.4 software and searches were done with Mascot 2.5 against forward and decoy Human-HHV4 Uniprot database (88,559 entries) with the following parameters: precursor mass tolerance [mass spectrum (MS)] 30 ppm, OT-MS/MS 0.06 Da, IT-MS/MS 0.6 Da; two miss cleavages; static modifications: carbamidomethyl (C), variable modifications: oxidation (M),

deamidated (NQ), acetyl N-terminal protein. Forward/decoy searches were used for false discovery rate (FDR) estimation (FDR 1%). Peak lists were generated.

Following data review, coupled with critical information (i.e. the protein's subcellular location and cellular functions) from UNIPROT (*UniProt Consortium, 2015*), candidates were shortlisted for further validation. The parasite culture suspensions were incubated with CSMT and antibodies against the shortlisted proteins (*Supplementary file 3*) and tested in rosetting assay at a final concentration of 25 µg/ml.

## IGFBP7 and rosetting

The parasite culture suspension was divided into three groups, one served as control, the second group was added with recombinant human IGFBP7 (final concentration 100 ng/ml), and the third group was mixed with CSMT (diluted with culture medium to make the CSMT 'working concentration' equivalent to that of $1 \times 10^6$ cells). After 1 h of incubation under in vitro cultivation conditions, rosetting assay was conducted. Separately, a portion of the IGFBP7 suspension was heat-denatured at 95°C for 1 h, prior to use in rosetting assay.

Parasite cultures were incubated with IGFBP7 (working concentrations 0–25,000 ng/ml) prior to rosetting assay. Time course experiments were performed with laboratory-adapted *P. falciparum* lines (3D7, MKK183, FVT402, FVT201, WPP3065) incubated with IGFBP7 (100 ng/ml). Rosetting assay was conducted after 5 min, using 7 µl of the suspension. The remaining suspension was kept back into the incubator. Rosetting assay was repeated at 5 min-intervals until 1 h-post-IGFBP7 exposure. Reversibility of IGFBP7-mediated rosette-stimulation was also tested. Parasite lines (3D7, MKK183, FVT402, FVT201, WPP3065) were incubated with IGFBP7 (100 ng/ml) and rosetting assay was conducted with 7 µl of the suspension. The remaining suspension was centrifuged at 1500 g for 5 min. Supernatant was removed, and the pellet was washed three times with culture medium, followed by re-suspension with culture medium. Five minutes later, 7 µl of the suspension was taken for rosetting assay, subsequently repeated at 5 min-intervals up to 1 h post-IGFBP7 removal.

## Identification of rosetting ligands and receptors that interact with IGFBP7

Magnetic activated cell sorter (MACS)-sorted late stage-IRBC (purity ≥95%) were trypsinized at different working concentrations. The first group was mixed with enzyme trypsin (final trypsin concentration of 10 µg/ml), and the second was mixed with enzyme trypsin (final trypsin concentration of 1 mg/ml). The third served as an untreated control. The cells were incubated at 37°C for 30 min. After that, the cells were washed with serum-enriched medium three times. Each group was incubated with or without IGFBP7 (100 ng/ml) prior to rosetting assay. *P. falciparum* line CS2 deficient of SBP1 [(SBP1-KO-CS2), which lacks *P. falciparum* erythrocyte membrane protein 1 (PFEMP1) on its IRBC surface] and its wild type (CS2-WT) counterpart were cultured as described (*Maier et al., 2007*; *Chan et al., 2016*). Their rosetting rates post-incubation with IGFBP7 at different concentrations were determined. *P. falciparum* clones NF54 VAR2CSA_WT and NF54_T934D (cannot express PfEMP1 variant VAR2CSA on IRBC surface [*Dorin-Semblat et al., 2019*]) were cultivated. Experiments were conducted when parasite population reached late stages. For each parasite line, two conditions were applied; one was incubated with IGFBP7 (100 ng/ml), whereas the other acted as a IGFBP7-free control. Rosetting assay was conducted afterwards. Five replicates were conducted for each experiment setting. In a separate experiment, a laboratory-adapted clinical isolate from Thai-Burmese border (NHP1106) was cultivated and staging of parasites was tightly synchronized. The experiment was conducted when the parasite population reached late rings (~hour 16–26). Two settings were prepared; one was incubated with IGFBP7 (100 ng/ml) and the other acted as a IGFBP7-free control. One hour of incubation under in vitro cultivation conditions was done prior to rosetting assay. Nine replicates (across three cycles of cultivations) were conducted.

The role of heparan sulfate (HS) in IGFBP7-mediated rosetting was also tested. URBC (blood group O) were treated with heparinase I (final working concentration of 25 µg/ml) or heparinase III (final working concentration of 25 µg/ml), with the untreated URBC served as control. The enzyme-erythrocyte mixtures and the untreated controls were incubated at 37°C for 30 min. After that, the suspension was centrifuged to remove supernatant. The treated erythrocytes were washed with 20% human serum enriched-culture medium three times. Subsequently, cells were suspended in plain

culture medium. The prepared cells were kept at 4°C until use within 1 week. Late stage-IRBC were concentrated with MACS. The IRBC (IRBC purity: 90–96%) were divided into three groups, each mixed with the control, heparinase I-treated and heparinase III-treated URBC, respectively.

The roles of complement receptor 1 (CR1/CD35) and A/B blood antigens were also investigated. Recruited isolates were matured in vitro, subsequently divided into four groups. One group served as the control, another group was added with rhIGFBP7 (final concentration 100 ng/ml). The third group was added with mouse anti-human CR1 (CD35) $IgG_1$ (final concentration 25 μg/ml), whereas the fourth group was added with rhIGFP7 (final concentration 100 ng/ml) and mouse anti-human CR1 $IgG_1$ (final concentration 25 μg/ml). Rosetting assay was conducted after the incubation. Separately, the late stage-IRBCs were sorted with MACS. The sorted cells were divided into four groups, each to be mixed with URBCs of A, B, O, and AB groups, respectively. Each of the cell mixture groups was further divided into two groups, where rhIGFBP7 (final concentration 100 ng/ml) was added into one group and the other group served as control. Culture media enriched with 20% AB serum were used. The experiment was repeated with CSMT replacing rhIGFBP7.

## Identification of serum-derived co-mediators in IGFBP7-mediated rosetting

An aliquot of human serum used for culture medium preparation was filtered with cellulose acetate syringe filter (pore size 0.45 μm). The filtered fraction was used to prepare 20% filtered serum-enriched RPMI1640 medium. Packed erythrocytes from cultures (*P. falciparum* laboratory-adapted lines: 3D7, CS2-WT, FVT201, MKK183, WPP3065) were divided into two groups. The first group was suspended with 20% complete human serum-enriched RPMI 1640 (denoted as 'human serum' group). The second group was suspended with the 20% filtered human AB serum-enriched medium (denoted as 'filtered human serum' group). Culture was further incubated with or without IGFBP7 (100 ng/ml) before rosetting assessment.

## Role of von Willebrand factor (VWF) and thrombospondin-1 (TSP-1) in IGFBP7-mediated rosetting

Culture suspension of the laboratory-adapted *P. falciparum* lines (3D7, MKK183, NHP1106, WPP3065, WPP2803, NHP4770, FVT201, FVT402) was centrifuged, and the packed cells were divided into groups: IGFBP7-free, anti-VWF, IGFBP7, and IGFBP7 + anti-VWF groups. Rabbit anti-human VWF polyclonal IgG was used at a working concentration of 25 μg/ml, whereas IGFBP7 at 100 ng/ml was applied. Rosetting assay was done after incubation. In another experiment, the packed cells of parasite cultures (3D7, MKK183, NHP1106, WPP3065, WPP2803, NHP4770, FVT201, FVT402) were divided into two parts, one was suspended with 20% serum-enriched RPMI1640 whereas the other group was suspended with 2% serum-enriched RPMI 1640. Each group was further divided into four categories, that is control, IGFBP7, VWF, and IGFBP7 + VWF. The working concentration of IGFBP7 was 100 ng/ml. For rhVWF (referred to as VWF), final concentration of 1 IU/ml was used. Rosetting assay was conducted after incubation. In a separate experiment, the packed cells of cultures (3D7, MKK183, NHP1106, WPP3065, NHP4770, FVT201, FVT402) were suspended with 0.25% Albumax II (Alb)-enriched RPMI1640, and divided into seven groups, each incubated with different concentrations of VWF (0, 0.06, 0.125, 0.25, 0.5, 1.0, 2.0 IU/ml) prior to rosetting assay.

The antibody blocking experiment using anti-VWF was repeated using mouse anti-human TSP-1 $IgG_{2B}$ in place of the anti-VWF antibody. Subsequently, experiments were conducted using rhTSP-1 (referred as TSP-1). The parasite culture packed cells were washed with plain RPMI 1640 medium twice. Each parasite line was divided into 12 groups. Ten groups were suspended with 0.25% Albumax-enriched medium (referred to as 'Alb' in this experiment) and the remaining two groups were suspended with 20% serum-enriched medium (referred to as '20% serum' in this experiment). The groups were as follows: IGFBP7-free Alb (control), Alb + IGFBP7 (100 ng/ml), Alb + VWF (2IU/ml), Alb + 10 ng/1 TSP-110 (henceforth referred to as $TSP-1_{10}$), Alb + $TSP-1_{10}$ + IGFBP7, Alb + $TSP-1_{10}$ + IGFBP7 + VWF, Alb + 500 ng/ml TSP-1 (referred to as $TSP-1_{500}$), Alb + $TSP-1_{500}$ + IGFBP7, Alb + $TSP-1_{500}$ + IGFBP7 + VWF, Alb + $TSP-1_{500}$ + VWF, 20% serum, 20% serum + IGFBP7. The working concentrations of IGFBP7 and VWF used were 100 ng/ml and 2 IU/ml, respectively. Rosetting assay was conducted after incubation.

To quantitate VWF needed in IGFBP7-mediated rosetting, the parasite culture packed cells (3D7, MKK183, NHP1106, WPP3065, NHP4770, FVT201, FVT402) were washed with plain RPMI 1640 medium twice. Subsequently, the cells were suspended with 0.25% Albumax-RPMI. Each isolate was further divided into seven categories, each added with different concentrations of VWF (0, 0.125, 0.5, 2.0 IU/ml). All these groups were given IGFBP7 (working concentrations 100 ng/ml) and TSP-1 (10 ng/ml). Rosetting assay was conducted after incubation. In a separate experiment, the levels of IGFBP7 of media used in the experiments (20% human serum-enriched RPMI media, Albumax-enriched RPMI media and plain RPMI media) were quantitated using an ELISA kit, following instructions provided by the kit's manufacturer.

## IGFBP7 secretion quantification

Peripheral monocytes (CD14$^+$) were purified from blood collected from five healthy donors via the Ficoll concentration method, followed by CD14$^+$ bead purification. The purified cells were suspended in 10% FBS-enriched RPMI 1640 medium. Three wells of 96-well flat bottom microplate were allocated to cells harvested from each donor, where $1 \times 10^5$ cells were seeded into each well. One well served as plain control, whereas the other well was incubated with URBC, and the third one was incubated with the purified *P. falciparum* 3D7 IRBCs (monocyte: IRBC ratio = 1:1000). The cells were incubated at in vitro cultivation for 24 h. Subsequently, supernatants of the cultures were collected separately. During supernatant collection, care was taken to minimize uptake of sedimented cells (RBCs, lysed cell products, hemoglobin may interfere with ELISA). Human IGFBP7 Duo-Set ELISA kit was used to measure the IGFBP7 level in the supernatant of each experiment group using the manufacturer's protocol. Measurements were done with microplate reader Tecan i-Control (Tecan).

The steps were repeated on THP-1, with slight modifications, where five laboratory-adapted *P. falciparum* lines (3D7, CS2-WT, FVT201, MKK183, WPP3065) were recruited. The mature stage-IRBCs were purified, and these purified IRBCs were then added with URBCs to make cell mixtures of parasitemia 16%, 8%, 4%, 2%, 1%–0.5% and 0.25%. Packed cells of only URBCs (0% parasitemia) was used as control. The cells were added into the Lab-Tek 8-chamber-slides that were already seeded with respective cell lines ($1 \times 10^5$ cells per well), making cellular suspension of 1.5% hematocrit. The cell mixtures were incubated for 24 h under in vitro cultivation conditions. Subsequently, supernatant was collected for ELISA analysis. During supernatant collection, care was taken to minimize uptake of sedimented cells (RBCs, lysed cell products, hemoglobin etc. may interfere with ELISA). ELISA was conducted on the supernatant collected.

## Knockdown of IGFBP7 expression by THP-1 using an shRNA approach

THP-1 cells were thawed and cultured with RPMI1640 medium enriched with 10% FBS. A 96-well plate was used. Each recruited well was seeded with $1 \times 10^4$ cells. For each well, 110 µl of medium and hexadimethrine bromide (final concentration 8 µg/ml; to enhance transduction) were added. Lentiviral transduction particles to knockdown expression IGFBP7 (hPGK-Puro_CMV-tGFP; SHCLNV-NM_001553) were added (MOI 3) based on formulas provided in the kit's user guide. On the following day, the media containing lentiviral particles were removed, and replaced with fresh medium. The next day, the transduced cells were cultivated with puromycin-added medium (working concentration 3 µg/ml) for selection. A small aliquot of the cells was examined with an epifluorescence microscope to check the GFP expression, which could be used for cell sorting (*Supplementary files 8–10*). The cells were used as 'IGFBP7-knockdown (KD) THP-1' in subsequent experiments.

Late stage-IRBCs (*P. falciparum* lines 3D7, CS2-WT, FVT201, MKK183, WPP3065) were purified. The WT- and IGFBP7-KD THP-1 were used. For each cell type, two groups (each group contains five sets, each well contained $1 \times 10^4$ cells) were prepared. One was added with the purified IRBCs (THP-1 to IRBC ratio of 1: 1000) and the other was added with URBCs from five healthy donors (control). RPMI enriched with 1% serum (to keep the viability of cells long enough for the experiment while minimizing the confounding effect on the protein quantification by the IGFBP7-KD cells) was used. The cells were incubated for 18 h at in vitro cultivation conditions. The supernatant of the cells was collected. Care must be taken to avoid uptake of cell/cell debris. The supernatant was used for IGFBP7 quantitation using ELISA and subsequent experiments described below.

The parasite culture packed cells (*P. falciparum* lines 3D7, CS2-WT, FVT201, MKK183, WPP3065) were divided into four groups. The first well was exposed to $1\times$ PBS (negative control), the second group was exposed to 100 ng/ml IGFBP7 (positive control), the third group was added with similar volume of culture supernatant collected from the IGFBP7-KD-THP-1 exposed to URBCs (CSKD-U), and the fourth group was added with culture supernatant collected from the IGFBP7-KD-THP-1 exposed to IRBCs (CSKD-I). The suspension was topped up with 20% serum-enriched medium and incubated for 1 h prior to rosetting assay.

The experiment was repeated with a control knockdown (knockdown of *Glycophorin C* [Gly C], a gene [cytogenetic location 2q14.3] that is different from *IGFBP7* gene [cytogenetic location 4q12] and with low expression levels in monocytes) and shRNA lentiviral vector, with slight modifications, where the approach of collecting CSMT/CSUT as described earlier was used. Quantification of IGFBP7 secretion was done. Besides, the culture supernatant was also used in rosetting assessment with laboratory-adapted *P. falciparum* line 3D7 coupled with use of anti-IGFBP7 antibody.

### Phagocytosis assessment

THP-1 was cultivated and expanded into three batches. For each batch of culture, $1 \times 10^6$ cells were incubated with IGFBP7 (working concentration 100 ng/ml) for 1 h under in vitro cultivation conditions. Another set of cells acted as IGFBP7-free control. Subsequently, Zymosan A (working concentration 10 μg/ml) was added to both sets of cells and incubated for another hour under in vitro cultivation conditions. Supravital staining with Giemsa was done for 15 min following this. Using a wet mount technique, the percentage of THP-1 cells which has engulfed Zymosan A was determined as phagocytosis rate by recruiting 1000 THP-1 cells. The experiment was repeated with the other two batches of THP-1 culture. And all the steps were repeated another two times using THP-1 cultures thawed at different time points. *P. falciparum* lines (3D7, MKK183, FVT402, FVT201, CS2_WT) were incubated with or without IGFBP7 in serum-enriched medium prior to rosetting assay. Subsequently, THP-1 were added. Using a wet mount technique, the IRBC phagocytosis rates were determined following the same formula to determine phagocytosis rate in the Zymosan A experiment. Prior to this, an experiment to compare IRBC phagocytosis activity of THP-1 and peripheral monocytes was conducted with a *P. falciparum* line (NHP1106), THP-1 and CD14+ peripheral monocytes from healthy donors, using the steps described above (five biological replicates conducted).

### Statistical analyses

GraphPad Prism 7.0 was used for data analysis. For normally distributed data (Shapiro-Wilk normality tested), a paired t-test was conducted for pairwise comparison. Matched measurement comparison for non-normally distributed datasets was done using Friedman test with Dunn's multiple comparison test. To compare two sets of non-normally distributed data, a Mann-Whitney test was used. One-way ANOVA tests were conducted for grouped data set comparison. For a normally distributed dataset, Tukey's test was applied for multiple group comparisons. Dunnett's test was used to compare groups against a control. Two-way ANOVA was used to study the effect of multiple experiment conditions on rosetting in different parasite lines, each with different culture batches. Any p values < 0.05 were interpreted as statistically significant.

### Acknowledgements

The authors would like to express gratitude to the staff of SMRU who assisted in the management of this study. This study received financial support from the following funds: WCL, SWH, SKB, BM and LR were supported by core funding from A*STAR to SIgN. LR was also funded by A*STAR grant (JCO-DP BMSI/15–800006-SIGN). WCL was also funded by Open Fund- Young Individual Research Grant (OF-YIRG NMRC/OFYIRG/0070/2018) by the National Medical Research Council, Ministry of Health, Singapore. RMS was supported by Core funding from IMCB and Young Investigator Grant YIG 2015 (A*STAR). BR was funded by University of Otago, Dunedin, New Zealand Start-Up Grant. YL was supported by University of Malaya High Impact Research (HIR) Grant (UM.C/HIR/MOHE/MED/16) from Ministry of Higher Education, Malaysia. BM is supported by NUHS start-up funding (NUHSRO/2018/006/SU/01) and NUHS seed fund (NUHRO/2018/094/T1). SMRU is part of the Mahidol-Oxford University Research Unit, supported by the Wellcome Trust of the Great Britain. We

thank the HSA Singapore for the supply of healthy uninfected blood. We also thank the flow cytometry team of SIgN for the assistance provided in flow cytometry and cell sorting.

## Additional information

### Funding

| Funder | Grant reference number | Author |
|---|---|---|
| Agency for Science, Technology and Research | SIgN core funding | Wenn-Chyau Lee<br>Khairunnisa Ghaffar<br>Shanshan W Howland<br>Subhra Biswas<br>Benoit Malleret<br>Laurent Renia |
| Agency for Science, Technology and Research | JCO-DP BMSI/15-800006-SIGN | Laurent Renia |
| National Medical Research Council | OF-YIRG NMRC/OFYIRG/0070/2018 | Wenn-Chyau Lee |
| Ministry of Higher Education, Malaysia | University of Malaya High Impact Research (HIR) Grant UM.C/HIR/MOHE/MED/16 | Yee-Ling Lau |
| Agency for Science, Technology and Research | IMCB Core funding | Radoslaw Mikolaj Sobota |
| Agency for Science, Technology and Research | Young Investigator Grant YIG 2015 | Radoslaw Mikolaj Sobota |
| Wellcome Trust | SMRU is part of the Mahidol-Oxford University Research Unit | Cindy Chu<br>François Nosten |
| National University Health System | Start-up funding NUHSRO/2018/006/SU/01 | Benoit Malleret |
| National University Health System | Seed fund NUHRO/2018/094/T1 | Benoit Malleret |

The funders had no role in study design, data collection and interpretation, or the decision to submit the work for publication.

### Author contributions

Wenn-Chyau Lee, Conceptualization, Data curation, Formal analysis, Funding acquisition, Validation, Investigation, Methodology, Writing - original draft, Project administration, Writing - review and editing; Bruce Russell, Formal analysis, Supervision, Validation, Methodology; Radoslaw Mikolaj Sobota, Resources, Investigation, Methodology, Writing - original draft, Writing - review and editing; Khairunnisa Ghaffar, Validation, Investigation, Methodology, Writing - review and editing; Shanshan W Howland, Investigation, Methodology, Writing - review and editing; Zi Xin Wong, Investigation, Writing - review and editing; Alexander G Maier, Dominique Dorin-Semblat, Subhra Biswas, Benoit Gamain, Resources, Methodology, Writing - review and editing; Yee-Ling Lau, Cindy Chu, François Nosten, Resources, Project administration, Writing - review and editing; Benoit Malleret, Resources, Investigation, Methodology, Writing - review and editing; Laurent Renia, Conceptualization, Resources, Data curation, Formal analysis, Supervision, Funding acquisition, Validation, Investigation, Project administration, Writing - review and editing

### Author ORCIDs

Wenn-Chyau Lee (iD) https://orcid.org/0000-0002-7324-5792
François Nosten (iD) http://orcid.org/0000-0002-7951-0745
Laurent Renia (iD) https://orcid.org/0000-0003-0349-1557

### Decision letter and Author response

Decision letter https://doi.org/10.7554/eLife.51546.sa1

Author response https://doi.org/10.7554/eLife.51546.sa2

## Additional files

### Supplementary files

• Supplementary file 1. Key resources table.

• Supplementary file 2. 694 components yielded from mass spectrometry on the aqueous fraction (molecular size ≤30 kDa) of CSMT.

• Supplementary file 3. Shortlisted candidates from a list of 694 compounds identified by mass spectrometry.

• Supplementary file 4. Recruited *P. falciparum* clinical isolates from the Thai-Burmese Border.

• Supplementary file 5. Recruited *P. vivax* clinical isolates from the Thai-Burmese Border.

• Supplementary file 6. Experiment flow. Flow chart showing the experiments done in the project, along with the number of samples recruited for each experiment.

• Supplementary file 7. Method comparison for rosetting assay. (A) Plot of rosetting rates obtained from recruited *P. falciparum* lines (n = 5) using different wet mount methods, with insets underneath the x-axis showing rosettes visualized by respective methods [immersion oil (1000×) magnification, scale bars represent 10 μm]. Pictures of unstained and Giemsa-wet mounts were taken using a light microscope Olympus BX43, whereas pictures of the acridine orange-wet mount were taken on an epifluorescence microscope Nikon TS100. One-way ANOVA with Tukey's test: unstained vs. Giemsa: p=0.9517. Acridine orange vs. unstained p>0.9999. Acridine orange vs. Giemsa: p=0.9809. (B) Changes of rosetting rates by IGFBP7 collected using different rosetting assays. Dotted lines were used to show read ups collected from different methods on the same sample. Dataset Giemsa did not pass normality test (Shapiro-Wilk normality test). Friedman with Dunn's test: unstained vs. Giemsa: p=0.3415; unstained vs. acridine orange: p=0.6177; Giemsa vs. acridine orange: p>0.9999, that is there was no significant difference between the methods used. n.s. not significant.

• Supplementary file 8. Profiling of THP-1 cell population prior to green fluorescent protein (GFP)-based sorting post-shRNA transduction. This is the profile of THP-1_WT as GFP-free control for cell sorting.

• Supplementary file 9. Profiling of IGFBP-KD THP-1 cell population prior to green fluorescent protein (GFP)-based sorting post-shRNA transduction.

• Supplementary file 10. Profiling of GlyC-KD THP-1 cell population prior to green fluorescent protein (GFP)-based sorting post-shRNA transduction.

• Transparent reporting form

### Data availability

All sample/data information in this study are included in the manuscript and supporting files (Supplementary files, Source data files). Of note, data represented as bar graphs are provided as source data tables (5 sets): Figure 1—source data 1; Figure 5—source data 1; Figure 5—source data 2; Figure 8—source data 1; Figure 8—source data 2.

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
