## [Decision Letter]

**Acceptance summary:**

Parasites elaborate many evasion mechanisms to avoid elimination by the host. One such mechanism, presumed to be used by the *Plasmodium* parasite, is rosetting. This mechanism involves aggregation of red blood cells around red blood cells infected with *Plasmodium falciparum* or *Plasmodium vivax*, two infectious parasites that cause severe malaria. In this manuscript, Lee and colleagues describe a novel type of rosetting and the authors name it type II rosetting. This novel rosetting is distinct from the type I rosetting in that it is enhanced by the addition of monocytes and is mediated by the release IGFBP7. Two other serum factors, the Von Willebrand Factor and Thrombospondin-1, are also required to mediate the type II rosetting. The functional importance of type II rosetting is its immune-evasion as it might hinder phagocytic uptake of infected red cells by macrophages. The details of the mechanism of rosetting are potentially crucial for aiding the development of vaccines and other immune-prophylactic approaches to eliminate *Plasmodium* parasites.

**Decision letter after peer review:**

[Editors’ note: the authors were asked to provide a plan for revisions before the editors issued a final decision. What follows is the editors’ letter requesting such plan.]

Thank you for sending your article entitled "*Plasmodium*-infected erythrocytes induce secretion of IGFBP7 to form type II rosettes and escape phagocytosis" for peer review at *eLife*. Your article is being evaluated by three peer reviewers, and the evaluation is being overseen by Dominique Soldati-Favre as the Senior Editor. The reviewers have opted to remain anonymous.

Given the list of essential revisions, including new experiments, the editors and reviewers invite you to respond with an action plan and timetable for the completion of the additional work.

Lee and colleagues describe a novel type of rosetting, called type II rosetting of *Plasmodium (P. falciparum* and *P. vivax*) infected red blood cells (IRBC) with uninfected red blood cells (URBC). Although rosetting has not been demonstrated in vivo (as it may be rather difficult), these authors perform series of experiments demonstrating a novel pathway/mechanism of this interesting in vitro phenomenon ascribed to *Plasmodium* IRBC. This type II rosetting is distinct from the type I rosetting in that it is enhanced by the addition of monocytes and both *P. falciparum* and *P. vivax* reveal an increased rate of rosetting. The increased rate of rosetting stems from a factor, namely IGFBP7, released during interaction between monocytes and IRBC. In addition, two other serum factors, the Von Willebrand Factor and Thrombospondin-1, are also required for the IGFBP-7-mediated rosetting. An interesting aspect of this study was the demonstration of a functional importance of type II rosetting as it may aid the parasite in immune-evasion in that rosetting hinders phagocytosis of IRBC.

The following is a rather long "list" of key concerns with both the experimental approaches and data interpretation.

1) The observation that IGFB7, released from monocytes and monocyte cell line THP-1, mediates rosetting is based on the interpretation of the data shown in Figure 3 (neutralizing antibody) and gene knockdown experiments (Figure 8). As regards the experiment with neutralizing antibody, it appears that a proper control is missing. The authors included isotype controls but do not state whether controls were at concentrations similar to anti-IGFB7 Ab. Also, the IgG control is against NID1 rather than being a nonspecific IgG. It might be helpful to show rosetting without CSMT in the figure as well to provide baseline. It is possible that the differences in Abs effect are not due to blocking of specific receptors but rather that they reflect differences in properties of Abs.

2) IGFB7 was identified as a factor that mediates rosetting, but it is not clear whether IGFB7 exhibit similar heat sensitivities as did the activity from the supernatant (Figure 1). The authors did not conduct any experiments that would show if the concentrations of IGFB7 found in the supernatant of THP1 cells are similar to those that mediate rosetting.

Figure 1B: autologous leukocytes are added. Why is it important with autologous cells? If the rosetting was as dependent as the authors try to state on IGFBP7, it should not make a difference whether it is autologous or heterologous cells?

Figure 1C: Are these autologous cells?

Figure 1C: 3 donors and 4 parasites should give 12 data points but only 10 are plotted.

Figure 1D lacks statistical testing.

3) Another concern regards the second major conclusion, namely that Von Willebrand Factor and Thrombospondin-1 are required for the effects of recombinant IGFB7 on rosetting. The involvement of these two factors must occur when they are added at saturating concentrations. Otherwise each factor might be effective through the same mechanism. Thus, a dose-response relationship needs to be demonstrated with Von Willebrand Factor and Thrombospondin and rosetting.

Throughout the manuscript the authors talk about "serum factors". But the only reference I can find when it comes to collection of blood, is lithium heparin. That means *plasma*, not *serum*. So is this paper actually about plasma factors?

Figure 3: In general, *P. vivax* generates a lot more rosetting than *P. falciparum*. What is the explanation for this?

4) In experiments examining enzyme sensitivity (trypsin in Figure 5 and heparinase in Figure 6), the authors do not remove the enzyme before testing for rosetting. Also, the authors report the effects of enzyme concentration rather than enzyme activity. The enzymes used are not pure and thus nonspecific effects must be considered.

5) The authors use a wide range of parasite lines and isolates from patients and there is a significant heterogeneity in the results obtained. The rational for the use of the different lab isolates needs to explained better. Also it should be made clear that clinical isolates most likely are clonal.

6) The VAR2CSA expressing parasite – can the authors confirm that it expressed VAR2CSA? What about CS2_WT?

7) In Figure 5D, the clinical isolate has very low rosetting as compared to many of the other isolates used. Can the authors suggest why this may be the case?

8) The VAR2CSA expressing line is particularly poor in rosetting. This suggests that levels of rosetting may be impacted by the PfEMP1 expressed. It would be interesting if the authors could provide some insights into this. For example what is unique about FVT402 as compared to 3D7 or MK183?

9) The authors propose that PfEMP1 is the parasite ligand that mediates this rosetting via TSP1. However, it is clear that the levels of rosetting varies between isolates and lines.

10) In Figure 6 the authors show that rosetting is impacted of in clinical isolates and is effected by anti-CR1 me. This result is somewhat surprising, as it would suggest that all the parasites express a CR1 binding PfEMP1. Isn't so that only a subset of PfEMP1 that bind CR1? Can the authors explain this?

11) Figure 7 and subsection “Serum-derived co-factors in IGFBP7-mediated rosetting”, the authors appear to state that large proteins were removed with a 0.45 micron filter. The statement is inaccurate. It is more likely the authors are removing large protein aggregates.

12) The experiment depicted in Figure 8 is missing controls. Lentiviral vector mediated knockdowns experiments require controls for off-target effects. It is necessary to have two shRNA vectors targeting different sequences on the same gene. The authors present only one. There may also be some confusion regarding the technique as subsection “Knockdown of IGFBP7 expression by THP-1 using RNAi transduction” states that siRNA was used and the correct term should be shRNA.

13) In an attempt to be concise and brief, the authors provide very little experimental details – particularly concerning how an experiment addresses a particular question in the main text. This makes it very difficult for someone that is not an expert in the field to understand the work. While the Materials and methods section provides all the relevant information, it is very detailed and therefore makes it difficult and cumbersome to find the relevant information. For example "using Zymosan A" for many readers this would not provide the relevant information to interpret the results. Other concerns arose regarding clarity throughout the manuscript. Below are some examples that seem confusing. When examining whether a cell-free supernatant contains a biological active substance, serial dilutions must be examined and the dilution resulting in 50% activity is then used for chemical characterization. If the factor is highly concentrated in supernatant, loss due to temperature, for example, might not be observed. Please revise the subsection “Effect of THP-1 culture supernatants and supernatant fractionation”, as it is confusing.

14) Finally, there is an objection to the characterization of the rosetting studied here as "type II" rosetting and distinguishing it from "type I" rosetting. In the Introduction section the authors write, "We further showed that IGFBP7-mediated rosetting was different from the previously described rosetting (defined here as type I rosetting), where it (we referred to as type II rosetting) required additional serum factors to occur, in addition to the interaction between the parasite-derived ligand on IRBC surface and the receptor on the surface of URBC." The requirement for serum factors in rosetting has been described previously. For example, Stevenson and colleagues have made extensive studies of α-2-macroglobulin, which can induce rosetting on its own, while other proteins need several factors together to be efficient. Would the authors reconsider this point, and simply call rosetting "type II, simply, rosetting?

[Editors’ note: after receiving the revision plan, further clarifications were requested prior to acceptance, as described below.]

Thank you for submitting a revision plan for your article entitled "*Plasmodium*-infected erythrocytes induce secretion of IGFBP7 to form type II rosettes and escape phagocytosis". We would like to invite you to submit a revised manuscript, bearing the following additional comments in mind:

The comment from the authors that they have actually been using serum and not plasma, is confusing and makes me doubt the results about von Willebrand factor.

a) When serum is formed, blood is coagulated. Platelets are activated, coagulation factors activated and von Willebrand factor is used. So how much vWf is there actually left in serum? Normally, when vWf is measured in humans, it is always measured in plasma. Often in the literature, people have difficulties separating serum and plasma, as for example in Ogawa et al., 2001 where the title contains "serum" but when I read in the Materials and methods what has actually been used, it is plasma. Also in the paper by Kastritis et al., 2016, they talk about "Serum levels of vWf" in the title but when I read the Materials and methods, they have used HemosIL which requires plasma to be able to function.

The authors have in this current paper not shown that there is any vWf in their serum, they just claim that the effect seen is because an antibody against vWf has an effect. And that antibody (rabbit polyclonal 6994) recognizes both bovine and human vWf. So how do we know that the antibody is not crossreacting with something else in the assay?

b) They have been using Albumax II, which contains <1% IgG. In Albumax I there is less than 0.1% IgG, which means that there is probably almost 1% IgG in Albumax II. Are there also other bigger proteins like von Willebrand factor? Albumax is made from bovine serum. If there should still be some vWf in human serum, which I from the above reasoning doubt, it could also be there in the Albumax.

So in conclusion, if the authors are going to continue to claim that there is an effect because of vWf, I would like to see it measured in their culture medium.

---

## [Author Response]

[Editors’ note: the authors’ plan for revisions was approved and the authors made a formal revised submission.]The following is a rather long "list" of key concerns with both the experimental approaches and data interpretation.1) The observation that IGFB7, released from monocytes and monocyte cell line THP-1, mediates rosetting is based on the interpretation of the data shown in Figure 3 (neutralizing antibody) and gene knockdown experiments (Figure 8). As regards the experiment with neutralizing antibody, it appears that a proper control is missing. The authors included isotype controls but do not state whether controls were at concentrations similar to anti-IGFB7 Ab. Also, the IgG control is against NID1 rather than being a nonspecific IgG. It might be helpful to show rosetting without CSMT in the figure as well to provide baseline. It is possible that the differences in Abs effect are not due to blocking of specific receptors but rather that they reflect differences in properties of Abs.

All the antibodies used in the work were at working concentration of 25 µg/ml, including the isotype controls as well. We apologize for overlooking this matter. We have added the information accordingly:

“…The specificity of rosette-inhibition by the antibodies was validated with experiments using antibody isotype controls at the same working concentration (25 µg/ml) (Figure 7—figure supplement 1A).”

The isotype controls were included to rule out non-specificity of antibodies. As shown in our Figure 7—figure supplement 1A, the presence of these antibody isotype controls did not affect the ability of IGFBP7 to induce more rosetting, and presence of these antibody isotypes alone exerted no significant changes on the rosetting rates as well. The usage of anti-NID1 in the experiment was also acting like an internal control for the experiment due to its negative effect.

We have added the rosetting rates without CSMT (control) in all graphs of Figure 3.

We disagree that the effect seen was not due to the specific blocking of receptors, since we have included the right types of antibody isotypes in the experiments. Kindly refer to Figure 7—figure supplement 1A for this, and the key resources table for the antibodies we used in this study.

2) IGFB7 was identified as a factor that mediates rosetting, but it is not clear whether IGFB7 exhibit similar heat sensitivities as did the activity from the supernatant (Figure 1). The authors did not conduct any experiments that would show if the concentrations of IGFB7 found in the supernatant of THP1 cells are similar to those that mediate rosetting.

Experiments in Figure 2D (the one involved heat treatment of culture supernatant) revealed that heat treatment lowered but not completely alleviated the rosette-stimulatory effect of the supernatant, demonstrating involvement of multiple players in rosette-stimulation. We did not try to imply that IGFBP7 is heat-stable. It was a process of deciphering the players of rosette-stimulation from the scratch. We performed a heat treatment experiment (Figures 4E-F) to show that the rosette-stimulating effect of our recombinant IGFBP7 suspension is not attributed to potential non-protein contaminant of the product, which also implies that the protein is not heat-stable. This also fits the picture (Figure 2D) that there is such a big difference in rosette-stimulation (P = 0.0009) between the unheated and heated compartments, as IGFBP7 is the major player in this phenomenon.

We did perform quantitation of IGFBP7 secreted by THP-1 into culture supernatant (Figure 8B). Exposure of parasites to 1x 10^5^ THP-1 cells for 24 hours was adequate to stimulate these cells to secrete more IGFBP7 (from baseline of ~10 ng/ml to ~20 ng/ml). The culture supernatant that we used for rosetting assay was of 1x 10^6^ cells.

Figure 1 B: autologous leukocytes are added. Why is it important with autologous cells? If the rosetting was as dependent as the authors try to state on IGFBP7, it should not make a difference whether it is autologous or heterologous cells?

At the beginning part of the experiment, we did not know about IGFBP7 at all, and tried our best to design experiments that minimized confounding factors as good as possible. Therefore, we used autologous cells. But subsequently, we used cells from healthy donors (Figure 1C), and still saw the rosette-increment. Hence, the requirement of autologous cells was dropped thereafter.

Figure 1C: Are these autologous cells?Figure 1C: 3 donors and 4 parasites should give 12 data points but only 10 are plotted.

3 donors were used for 3 parasites (9 plots). Not long after the experiment, we received cells from another donor, and on that day, a coworker accidentally thawed a *P. falciparum* clinical isolate [RDM00036, as stated in Materials and methods] that was not usable for her vivax malaria research. Hence, we recruited the donor’s cells and the unwanted *P. falciparum*, and had our 10^th^ plot in the graph.

Figure 1D lacks statistical testing.

We did perform Two-way ANOVA with Tukey’s multiple comparison test for this. It was too messy to plot asterisks across the bars. Hence we put the asterisks on the figure legends, since all parasites showed the similar degree of significance (within the same asterisk category) for all the comparisons.

3) Another concern regards the second major conclusion, namely that Von Willebrand Factor and Thrombospondin-1 are required for the effects of recombinant IGFB7 on rosetting. The involvement of these two factors must occur when they are added at saturating concentrations. Otherwise each factor might be effective through the same mechanism. Thus, a dose-response relationship needs to be demonstrated with Von Willebrand Factor and Thrombospondin and rosetting.

We did perform such dose-effect experiments for VWF (Figure 7D) and TSP-1 (Figure 7F) by fixing the dose of IGFBP7 at 100 ng/ml. In fact, it was through experiment in Figure 7D that we realized that there was another serum-derived player involved in this. And via experiment Figure 7F, by fixing the working concentrations of IGFBP7 and VWF, we tested effect of TSP-1 at 10 ng/ml and 500 ng/ml. We decided the working concentrations of VWF and TSP-1 based on literature review. For healthy individuals, the serum VWF levels range from 0.48 to 1.24 IU/ml, whereas individual having underlying pathological conditions could have much higher levels of VWF (> 300 IU/ml) (Terpos et al., 2013; Kasritis et al., 2016). Therefore, we set the working concentrations of VWF at 2 IU/ml. The range of reported serum TSP-1 concentrations in healthy individuals vary greatly, with one study reported a range of 0-12060 ng/ml (Rouanne et al., 2016). The reported serum TSP-1 levels in healthy individuals by another study varied according to obesity status of the individuals, where non-obese individuals recorded serum TSP-1 levels of 162.7 ± 34.5 ng/ml and obese individuals recorded serum TSP-1 levels of 220.9 ± 46.7 ng/ml (Liu et al., 2015). Prior to experiment in Figure 7F, we already found that the third serum-derived player was only required in low concentration, where even a 2% serum-enriched RPMI1640 medium was adequate to support IGFBP7-mediated rosetting. With this information, two working concentrations of TSP-1 were used in the experiment, i.e. 10 ng/ml and 500 ng/ml.

*Throughout the manuscript the authors talk about "serum factors". But the only reference I can find when it comes to collection of blood, is lithium heparin. That means* plasma*, not* serum*. So is this paper actually about plasma factors?*

We collected the parasite-containing packed blood cells that should not be coagulated with lithium heparin-anticoagulant vacutainers. But for culture media (same as experiment media) preparation, we use plain tubes, which allow the blood to coagulate, then the *serum* is collected and used as enrichment for RPMI1640. Subsequently, when we replaced the serum with Albumax as enrichment for the media, the IGFBP7-mediated rosette-stimulation was not seen. Hence, “serum factor” is the suitable word to be used in this context.

Figure 3: In general, P. vivax generates a lot more rosetting than *P. falciparum*. What is the explanation for this?

It seems like this high rosetting property is common among the *P. vivax* population, or at least those in Thailand. It was reported by Chotivanich et al. two decades ago (Chotivanich et al., 1998), and by our group a few years ago (Lee et al., 2014). We do not know the reason behind this. It may be a geographical origin-specific trait, but it is likely to be a common trait in *P. vivax*, since rosetting has been commonly found in *P. vivax* from the New World as well (Marin-Menendez et al., 2013, PLoS Negl Trop Dis 7:e2155).

4) In experiments examining enzyme sensitivity (trypsin in Figure 5 and heparinase in Figure 6), the authors do not remove the enzyme before testing for rosetting. Also, the authors report the effects of enzyme concentration rather than enzyme activity. The enzymes used are not pure and thus nonspecific effects must be considered.

We disagree with the reviewer. We mentioned about the washing steps in the Materials and methods. The rosetting assay was conducted under conditions free of these enzymes. The reporting of trypsin concentration in works associated with rosetting ligands of *Plasmodium* spp. has always been done in such manner (Crabb et al., 1997, Cell 89:287). Since the enzyme solutions in market are not 100% Trypsin, presenting the values in weight/volume based on calculation after referring to the product’s information sheet should be the best way forward for presentation. We understand that there may be underlying risk of non-specificity when using trypsin solutions. Hence, we used other approaches to decipher the rosetting ligands involved as well.

5) The authors use a wide range of parasite lines and isolates from patients and there is a significant heterogeneity in the results obtained. The rational for the use of the different lab isolates needs to explained better. Also it should be made clear that clinical isolates most likely are clonal.

Our intention was to generate data that can represent the actual parasite population that we studied as close as possible, hence the use of clinical isolates. However, the number of cases decreased rapidly during the course of this study. Many mainstream parasite lab lines that have been cultivated for many years in Albumax-enriched media have lost their rosetting ability. Although some of them managed to show rosetting after long period of cultivation in serum-enriched media (e.g. 3D7, CS2), their rosetting rates were always on the lower end without purification of rosette-forming subpopulation (which we do not want, as such purification may not reflect the actual population dynamics of the parasites) in each cycle of cultivation. Hence, we decided to adapt as many clinical isolates that we had into the in vitro cultivation conditions, and used them for the experiments in this study.

We do not know and we cannot assume, as the reviewer does, that the clinical isolate-derived cultures are clonal since we did not assess their level of clonality.

6) The VAR2CSA expressing parasite – can the authors confirm that it expressed VAR2CSA? What about CS2_WT?

For the VAR2CSA expressing parasite lines NF54 VAR2CSA_WT and NF54_T934D, sequencing was done to check the VAR2CSA integrity and the presence of expected mutations. Kindly refer to Figure 4 of a recently published paper (Dorin-Semblat et al., 2019) for more detail.

For CS2_WT, we performed a trypsin assay. The digestion pattern is the same as for Var2CSA (the antibody is anti-ATS) (Author response image 1). As this is actually part of another study, we do not wish to include this figure as the figure or supplemental figure for this manuscript.

7) In Figure 5D, the clinical isolate has very low rosetting as compared to many of the other isolates used. Can the authors suggest why this may be the case?

The experiment in Figure 5D was conducted on late ring stages, which, as explained in the manuscript, is the developmental stage where PfEMP1 is the only surface-expressed rosetting ligand. Although the surface expression is already initiated at this developmental stage, the surface expression of rosetting ligands will only reach optimal state hours later (late trophozoite stage). Hence, the rosetting rates of the IRBCs in this experiment (which used late ring stages, as compared to the rest of the experiments that used mature/late stages were lower.

8) The VAR2CSA expressing line is particularly poor in rosetting. This suggests that levels of rosetting may be impacted by the PfEMP1 expressed. It would be interesting if the authors could provide some insights into this. For example what is unique about FVT402 as compared to 3D7 or MK183?

We agree with the reviewers that this is an important and interesting aspect, given that 3D7 is a parasite line of African origin, whereas FVT402 and MKK183 are the Thai isolates. We believe this is beyond the scope of the current study, since compiling all this information into one manuscript will be too overwhelming and will distract from the core focus of this work: IGFBP7-mediated rosetting.

9) The authors propose that PfEMP1 is the parasite ligand that mediates this rosetting via TSP1. However, it is clear that the levels of rosetting varies between isolates and lines.

Different isolate and line may have different saturations/levels of IRBC surface-expressed PfEMP1. This may be one reason for the different rosetting rates seen. In addition, the sera that we used as media enrichment, as well as the erythrocytes that we used in cultivation were from different individuals. These may contribute to the variation seen. We tried our best to use consistent conditions and materials for the experiments through the years. However, these are the aspects that we cannot control. Nevertheless, it is important to emphasize that collectively, TSP-1 and VWF facilitate the IGFBP7-mediated rosetting in the isolates and lines tested.

10) In Figure 6 the authors show that rosetting is impacted of in clinical isolates and is effected by anti-CR1 me. This result is somewhat surprising, as it would suggest that all the parasites express a CR1 binding PfEMP1. Isn't so that only a subset of PfEMP1 that bind CR1? Can the authors explain this?

Not all recruited *P. falciparum* isolates showed big rosette-inhibition by the anti-CR1 antibodies. However, some showed drastic rosetting rate reduction by the antibody. Collectively, the dataset showed significant difference. Non-specific inhibition such as steric hindrance may be a possible reason. However, it is rather unlikely the case here, since our data from *P. vivax* isolates showed that the same antibody had insignificant effect on their rosetting rates. Our dataset pattern still fits into the statement that only a subset of PfEMP1 rosette via CR1 binding.

11) In Figure 7 and subsection “Serum-derived co-factors in IGFBP7-mediated rosetting”, the authors appear to state that large proteins were removed with a 0.45 micron filter. The statement is inaccurate. It is more likely the authors are removing large protein aggregates.

We thank the reviewer for this, this has been amended:

“All the experiments described above were performed using 20% human serum-enriched medium. However, serum filtration with a 0.45 μm filter abolished IGFBP7-induced rosetting (Figure 7A). These results indicated that other large-sized serum-derived protein aggregates or multimers might be needed for the IGFBP7-mediated rosetting effect.”

12) The experiment depicted in Figure 8 is missing controls. Lentiviral vector mediated knockdowns experiments require controls for off-target effects. It is necessary to have two shRNA vectors targeting different sequences on the same gene. The authors present only one. There may also be some confusion regarding the technique as subsection “Knockdown of IGFBP7 expression by THP-1 using RNAi transduction” states that siRNA was used and the correct term should be shRNA.

This has been amended:

“…THP-1 cells were transduced with shRNA lentiviral vectors specific…”

13) In an attempt to be concise and brief, the authors provide very little experimental details – particularly concerning how an experiment addresses a particular question in the main text. This makes it very difficult for someone that is not an expert in the field to understand the work. While the Materials and methods section provides all the relevant information, it is very detailed and therefore makes it difficult and cumbersome to find the relevant information. For example "using Zymosan A" for many readers this would not provide the relevant information to interpret the results. Other concerns arose regarding clarity throughout the manuscript. Below are some examples that seem confusing. When examining whether a cell-free supernatant contains a biological active substance, serial dilutions must be examined and the dilution resulting in 50% activity is then used for chemical characterization. If the factor is highly concentrated in supernatant, loss due to temperature, for example, might not be observed. Please revise the sentence in subsection “Effect of THP-1 culture supernatants and supernatant fractionation”, as it is confusing.

We have taken the advice and added more “bridges” to explain or justify the need for subsequent experiments that came after the previous ones.

We have added brief explanation for zymosan A:

“…We hypothesized that IGFBP7-mediated rosetting could be a strategy used by the parasites to avoid phagocytosis. To test this, we performed a control experiment using Zymosan A (a protein-carbohydrate complex prepared from yeast cell wall, commonly used in phagocytosis assays) and showed that IGFBP7 by itself did not inhibit the phagocytosis ability of THP-1 (Figure 8E).”

We have made changes:

“Fractionation of CSMT into aqueous and lipid fractions revealed that the rosette-stimulating factors were in the aqueous fraction (Figure 2A). Subsequently, we further fractionated the aqueous fraction into high and low molecular weight sub-fractions (with the cut-off of 30 kDa). Both aqueous sub-fractions induced rosetting of *P. falciparum* and *P. vivax* (Figures 2B and 2C), demonstrating that the rosetting stimulation was mediated by multiple secreted hydrophilic factors, predominantly of sizes ≤ 30 kDa (particularly for *P. falciparum*).”

14) Finally, there is an objection to the characterization of the rosetting studied here as "type II" rosetting and distinguishing it from "type I" rosetting. In the Introduction section the authors write, "We further showed that IGFBP7-mediated rosetting was different from the previously described rosetting (defined here as type I rosetting), where it (we referred to as type II rosetting) required additional serum factors to occur, in addition to the interaction between the parasite-derived ligand on IRBC surface and the receptor on the surface of URBC." The requirement for serum factors in rosetting has been described previously. For example, Stevenson and colleagues have made extensive studies of α-2-macroglobulin, which can induce rosetting on its own, while other proteins need several factors together to be efficient. Would the authors reconsider this point, and simply call rosetting "type II, simply, rosetting?

The reason we redefine rosettes as “type I” and “type II” is that the classical type I is the rosette involving direct interaction between a rosetting ligand and a rosetting receptor. However, the type II rosette involve other factors that bring the rosetting ligand and receptor together, without which the rosetting rates would be dropped to the baseline levels contributed by the type I rosettes. We would like to stress that we are not claiming ourselves as the discoverer of this phenomenon. Indeed, as discussed in the manuscript, complement factor d, as well as the macroglobulin brought up by the reviewer are rosette stimulators as well. However, these factors are neither ligand nor receptors, which by themselves can also form stable rosettes too. Hence, we think that it is important to differentiate between the two types of rosettes.

The comment from the authors that they have actually been using serum and not plasma, is confusing and makes me doubt the results about von Willebrand factor.a) When serum is formed, blood is coagulated. Platelets are activated, coagulation factors activated and von Willebrand factor is used. So how much VWF is there actually left in serum? Normally, when VWF is measured in humans, it is always measured in plasma. Often in the literature, people have difficulties separating serum and plasma, as for example in Ogawa et al., 2001 where the title contains "serum" but when I read in the Materials and methods what has actually been used, it is plasma. Also in the paper by Kastritis et al., 2016, they talk about "Serum levels of VWF" in the title but when I read the Materials and methods, they have used HemosIL which requires plasma to be able to function.The authors have in this current paper not shown that there is any VWF in their serum, they just claim that the effect seen is because an antibody against VWF has an effect. And that antibody (rabbit polyclonal 6994) recognizes both bovine and human VWF. So how do we know that the antibody is not crossreacting with something else in the assay?b) They have been using Albumax II, which contains <1% IgG. In Albumax I there is less than 0.1% IgG, which means that there is probably almost 1% IgG in Albumax II. Are there also other bigger proteins like von Willebrand factor? Albumax is made from bovine serum. If there should still be some VWF in human serum, which I from the above reasoning doubt, it could also be there in the Albumax.So in conclusion, if the authors are going to continue to claim that there is an effect because of VWF, I would like to see it measured in their culture medium.

We have quantitated the VWF levels of the media used in this study. We took sera samples of two of our main serum contributing sources and prepared the culture media. The results have been included as Figure 7—figure supplement 1B. The level of VWF in our serum-enriched media was of adequate levels (higher than 0.125 IU/ml) to support the IGFBP7-mediated rosetting. Albumax-enriched medium and plain medium showed VWF level (if any) that was below the detection limit of the ELISA kit (2.56 x 10^-5^ IU/ml). We have included this information in the text as well:

“…Lastly, we quantitated the amount of VWF needed to facilitate IGFBP7-mediated rosetting. When supplemented with IGFBP7 and TSP-1 in Albumax-supplemented medium, VWF as low as 0.125 IU/ml was sufficient to significantly increase rosetting rates, with an optimal increment attained at 0.5 IU/ml (Figure 7G). In fact, the 20% serum-enriched media that we used for this study contained VWF higher than 0.125 IU/ml (Figure 7—figure supplement 1B).”

We would like to take this opportunity to disagree with the comment that serum does not have VWF. In the booklet of the ELISA kit that we used, it also acknowledged serum as one of the suitable samples for VWF. Based on the suggested dilution, serum seems to be better than plasma to recover low level of VWF.

Moreover, quantification of VWF using plasma and serum has been compared, and the levels of VWF of plasma and serum showed no significant difference (P = 0.752) (Kovacevic et al., 2019. *Atherosclerosis, 290*, 31-36. doi: 10.1016/j.atherosclerosis.2019.09.003).

Control knock experiments in THP1

As promised in the previous round of rebuttal, we have also conducted a shRNA transduction to knock down an unrelated gene (glycophorin C) on THP-1 using the same lentiviral vectors with the same molecular backbone produced by the same company (Sigma) as a control. The expression of IGFBP7 by the GlyC-KD THP-1 was not affected by the knockdown of GlyC, and the cells secreted more IGFBP7 when exposed to the parasites.

In addition, we also quantitated the level of IGFBP7 in the culture supernatant of THP-1-WT, IGFBP7-KD-THP-1, and GlyC-KD-THP-1, and used them for rosetting assay on *P. falciparum*, along with usage of anti-human IGFBP7 antibody in the experiment. We have rewritten the section to include this in the text:

“THP-1 cells were transduced with shRNA lentiviral vectors specific for IGFBBP7 and vectors specific for an unrelated protein, glycophorin C. Non-transduced cells served as wild types (THP-1_WT). […] Importantly, the significant difference between culture supernatant groups “THP-1_WT_URBC” and “THP-1_WT_IRBC” (P = 0.0018), as well as “GlyC-KD_THP-1_URBC” and “GlyC-KD_THP-1_IRBC” (P = 0.0013), but not between “IGFBP7-KD_THP-1_URBC” and “IGFBP7-KD THP-1_IRBC” (P = 0.6011) strongly suggests that IGFBP7 may be one of the key secreted products by the monocytic cells in response to parasite exposure, whereas other factors involved in rosette-stimulation may be secreted at baseline levels with or without presence of the parasites (Figure 8—figure supplement 2B).”